# SELECTIVE CLASSIFICATION CAN MAGNIFY DISPARITIES ACROSS GROUPS

**Erik Jones,**[*] **Shiori Sagawa,**[*] **Pang Wei Koh,**[*] **Ananya Kumar & Percy Liang**
Department of Computer Science, Stanford University
{erjones,ssagawa,pangwei,ananya,pliang}@cs.stanford.edu

## ABSTRACT

Selective classification, in which models can abstain on uncertain predictions, is a natural approach to improving accuracy in settings where errors are costly but abstentions are manageable. In this paper, we find that while selective classification can improve average accuracies, it can simultaneously magnify existing accuracy disparities between various groups within a population, especially in the presence of spurious correlations. We observe this behavior consistently across five vision and NLP datasets. Surprisingly, increasing abstentions can even *decrease* accuracies on some groups. To better understand this phenomenon, we study the margin distribution, which captures the model's confidences over all predictions. For symmetric margin distributions, we prove that whether selective classification monotonically improves or worsens accuracy is fully determined by the accuracy at full coverage (i.e., without any abstentions) and whether the distribution satisfies a property we call left-log-concavity. Our analysis also shows that selective classification tends to magnify full-coverage accuracy disparities. Motivated by our analysis, we train distributionally-robust models that achieve similar full-coverage accuracies across groups and show that selective classification uniformly improves each group on these models. Altogether, our results suggest that selective classification should be used with care and underscore the importance of training models to perform equally well across groups at full coverage.

## 1 INTRODUCTION

Selective classification, in which models make predictions only when their confidence is above a threshold, is a natural approach when errors are costly but abstentions are manageable. For example, in medical and criminal justice applications, model mistakes can have serious consequences, whereas abstentions can be handled by backing off to the appropriate human experts. Prior work has shown that, across a broad array of applications, more confident predictions tend to be more accurate (Hanczar & Dougherty, 2008; Yu et al., 2011; Toplak et al., 2014; Mozannar & Sontag, 2020; Kamath et al., 2020). By varying the confidence threshold, we can select an appropriate trade-off between the abstention rate and the (selective) accuracy of the predictions made.

In this paper, we report a cautionary finding: while selective classification improves average accuracy, it can magnify existing accuracy disparities between various groups within a population, especially in the presence of spurious correlations. We observe this behavior across five vision and NLP datasets and two popular selective classification methods: softmax response (Cordella et al., 1995; Geifman & El-Yaniv, 2017) and Monte Carlo dropout (Gal & Ghahramani, 2016). Surprisingly, we find that increasing the abstention rate can even *decrease* accuracies on the groups that have lower accuracies at full coverage: on those groups, the models are not only wrong more frequently, but their confidence can actually be *anticorrelated* with whether they are correct. Even on datasets where selective classification improves accuracies across all groups, we find that it preferentially helps groups that already have high accuracies, further widening group disparities.

These group disparities are especially problematic in the same high-stakes areas where we might want to deploy selective classification, like medicine and criminal justice; there, poor performance on particular groups is already a significant issue (Chen et al., 2020; Hill, 2020). For example, we study a variant of CheXpert (Irvin et al., 2019), where the task is to predict if a patient has pleural

---

[*]Equal contribution

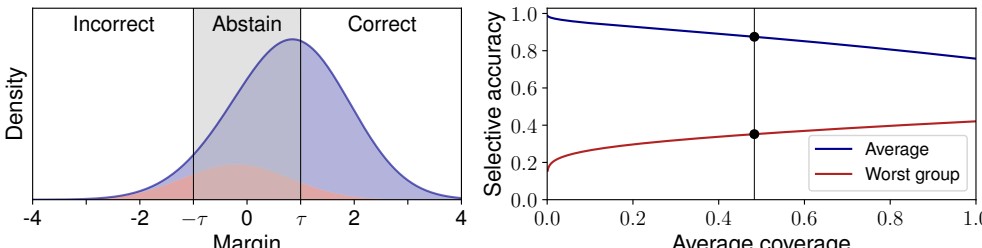

Figure 1: A selective classifier $(\hat{y}, \hat{c})$ makes a prediction $\hat{y}(x)$ on a point $x$ if its confidence $\hat{c}(x)$ in that prediction is larger than or equal to some threshold $\tau$. We assume the data comprises different groups each with their own data distribution, and that these group identities are not available to the selective classifier. In this figure, we show a classifier with low accuracy on a particular group (red), but high overall accuracy (blue). **Left**: The margin distributions overall (blue) and on the red group. The margin is defined as $\hat{c}(x)$ on correct predictions $(\hat{y}(x) = y)$ and $-\hat{c}(x)$ otherwise. For a threshold $\tau$, the selective classifier is thus incorrect on points with margin $\leq -\tau$; abstains on points with margin between $-\tau$ and $\tau$; and is correct on points with margin $\geq \tau$. **Right**: By varying $\tau$, we can plot the *accuracy-coverage curve*, where the coverage is the proportion of predicted points. As coverage decreases, the average (selective) accuracy increases, but the worst-group accuracy *decreases*. The black dots correspond to the threshold $\tau = 1$, which is shaded on the left.

effusion (fluid around the lung) from a chest x-ray. As these are commonly treated with chest tubes, models can latch onto this spurious correlation and fail on the group of patients with pleural effusion but not chest tubes or other support devices. However, this group is the most clinically relevant as it comprises potentially untreated and undiagnosed patients (Oakden-Rayner et al., 2020).

To better understand why selective classification can worsen accuracy and magnify disparities, we analyze the margin distribution, which captures the model's confidences across all predictions and determines which examples it abstains on at each threshold (Figure 1). We prove that when the margin distribution is symmetric, whether selective classification monotonically improves or worsens accuracy is fully determined by the accuracy at full coverage (i.e., without any abstentions) and whether the distribution satisfies a property we call left-log-concavity. To our knowledge, this is the first work to characterize whether selective classification (monotonically) helps or hurts accuracy in terms of the margin distribution, and to compare its relative effects on different groups.

Our analysis shows that selective classification tends to magnify accuracy disparities that are present at full coverage. Motivated by our analysis, we find that selective classification on group DRO models (Sagawa et al., 2020)—which achieve similar accuracies across groups at full coverage by using group annotations during training—uniformly improves group accuracies at lower coverages, substantially mitigating the disparities observed on standard models that are instead optimized for average accuracy. This approach is not a silver bullet: it relies on knowing group identities during training, which are not always available (Hashimoto et al., 2018). However, these results illustrate that closing disparities at full coverage can also mitigate disparities due to selective classification.

## 2 RELATED WORK

**Selective classification.** Abstaining when the model is uncertain is a classic idea (Chow, 1957; Hellman, 1970), and uncertainty estimation is an active area of research, from the popular approach of using softmax probabilities (Geifman & El-Yaniv, 2017) to more sophisticated methods using dropout (Gal & Ghahramani, 2016), ensembles (Lakshminarayanan et al., 2017), or training snapshots (Geifman et al., 2018). Others incorporate abstention into model training (Bartlett & Wegkamp, 2008; Geifman & El-Yaniv, 2019; Feng et al., 2019) and learn to abstain on examples human experts are more likely to get correct (Raghu et al., 2019; Mozannar & Sontag, 2020; De et al., 2020). Selective classification can also improve out-of-distribution accuracy (Pimentel et al., 2014; Hendrycks & Gimpel, 2017; Liang et al., 2018; Ovadia et al., 2019; Kamath et al., 2020). On the theoretical side, early work characterized optimal abstention rules given well-specified models (Chow, 1970; Hellman & Raviv, 1970), with more recent work on learning with perfect precision (El-Yaniv & Wiener, 2010; Khani et al., 2016) and guaranteed risk (Geifman & El-Yaniv, 2017). We build on this literature by establishing general conditions on the margin distribution for when selective classification helps, and importantly, by showing that it can magnify group disparities.

**Group disparities.** The problem of models performing poorly on some groups of data has been widely reported (e.g., Hovy & Søgaard (2015); Blodgett et al. (2016); Corbett-Davies et al. (2017); Tatman (2017); Hashimoto et al. (2018)). These disparities can arise when models latch onto spurious correlations, e.g., demographics (Buolamwini & Gebru, 2018; Borkan et al., 2019), image backgrounds (Ribeiro et al., 2016; Xiao et al., 2020), spurious clinical variables (Badgeley et al., 2019; Oakden-Rayner et al., 2020), or linguistic artifacts (Gururangan et al., 2018; McCoy et al., 2019). These disparities have implications for model robustness and equity, and mitigating them is an important open challenge (Dwork et al., 2012; Hardt et al., 2016; Kleinberg et al., 2017; Duchi et al., 2019; Sagawa et al., 2020). Our work shows that selective classification can exacerbate this problem and must therefore be used with care.

## 3 SETUP

A selective classifier takes in an input $x \in \mathcal{X}$ and either predicts a label $y \in \mathcal{Y}$ or abstains. We study standard confidence-based selective classifiers $(\hat{y}, \hat{c})$, where $\hat{y} : \mathcal{X} \to \mathcal{Y}$ outputs a prediction and $\hat{c} : \mathcal{X} \to \mathbb{R}_+$ outputs the model's confidence in that prediction. The selective classifier abstains on $x$ whenever its confidence $\hat{c}(x)$ is below some threshold $\tau$ and predicts $\hat{y}(x)$ otherwise.

**Data and training.** We consider a data distribution $\mathcal{D}$ over $\mathcal{X} \times \mathcal{Y} \times \mathcal{G}$, where $\mathcal{G} = \{1, 2, \ldots, k\}$ corresponds to a group variable that is *unobserved* by the model. We study the common setting where the model $\hat{y}$ is trained under full coverage (i.e., without taking into account any abstentions) and the confidence function $\hat{c}$ is then derived from the trained model, as described in the next paragraph. We will primarily consider models $\hat{y}$ trained by empirical risk minimization (i.e., to minimize the average training loss); in that setting, the group $g \in \mathcal{G}$ is never observed. In Section 7, we will consider the group DRO training algorithm (Sagawa et al., 2020), which observes $g$ at *training* time only. In both cases, $g$ is not observed at *test* time and the model thus does not take in $g$. This is a common assumption: e.g., we might want to ensure that a face recognition model has equal accuracies across genders, but the model only sees the photograph ($x$) and not the gender ($g$).

**Confidence.** We will primarily consider *softmax response* (SR) selective classifiers, which take $\hat{c}(x)$ to be the normalized logit of the predicted class. Formally, we consider models that estimate $\hat{p}(y \mid x)$ (e.g., through a softmax) and predict $\hat{y}(x) = \arg\max_{y \in \mathcal{Y}} \hat{p}(y \mid x)$, with the corresponding probability estimate $\hat{p}(\hat{y}(x) \mid x)$. For binary classifiers, we define the confidence $\hat{c}(x)$ as

$$\hat{c}(x) = \frac{1}{2} \log \left( \frac{\hat{p}(\hat{y}(x) \mid x)}{1 - \hat{p}(\hat{y}(x) \mid x)} \right). \tag{1}$$

This corresponds to a confidence of $\hat{c}(x) = 0$ when $\hat{p}(\hat{y}(x) \mid x) = 0.5$, i.e., the classifier is completely unsure of its prediction. We generalize this notion to multi-class classifiers in Section A.1. Softmax response is a popular technique applicable to neural networks and has been shown to improve average accuracies on a range of applications (Geifman & El-Yaniv, 2017). Other methods offer alternative ways of computing $\hat{c}(x)$; in Appendix B.1, we also run experiments where $\hat{c}(x)$ is obtained via Monte Carlo (MC) dropout (Gal & Ghahramani, 2016), with similar results.

**Metrics.** The performance of a selective classifier at a threshold $\tau$ is typically measured by its *average (selective) accuracy* on its predicted points, $\mathbb{P}[\hat{y}(x) = y \mid \hat{c}(x) \geq \tau]$, and its *average coverage*, i.e., the fraction of predicted points $\mathbb{P}[\hat{c}(x) \geq \tau]$. We call the average accuracy at threshold 0 the *full-coverage accuracy*, which corresponds to the standard notion of accuracy without any abstentions. We always use the term *accuracy* w.r.t. some threshold $\tau \geq 0$; where appropriate, we emphasize this by calling it *selective accuracy*, but we use these terms interchangeably in this paper.

Following convention, we evaluate models by varying $\tau$ and tracing out the *accuracy-coverage curve* (El-Yaniv & Wiener, 2010). As Figure 1 illustrates, this curve is fully determined by the distribution of the *margin*, which is $\hat{c}(x)$ on correct predictions ($\hat{y}(x) = y$) and $-\hat{c}(x)$ otherwise. We are also interested in evaluating performance on each group. For a group $g \in \mathcal{G}$, we compute its *group (selective) accuracy* by conditioning on the group, $\mathbb{P}[\hat{y}(x) = y \mid g, \hat{c}(x) \geq \tau]$, and we define *group coverage* analogously as $\mathbb{P}[\hat{c}(x) \geq \tau \mid g]$. We pay particular attention to the worst group under the model, i.e., the group $\arg\min_g \mathbb{P}[\hat{y}(x) = y \mid g]$ with the lowest accuracy at full coverage. In our setting, we only use group information to evaluate the model, which does not observe $g$ at training or test time; we relax this later when studying distributionally-robust models that use $g$ during training.

| Dataset | Modality | # examples | Prediction task | Spurious attributes $\mathcal{A}$ |
|---|---|---|---|---|
| CelebA | Photos | 202,599 | Hair color | Gender |
| CivilComments | Text | 448,000 | Toxicity | Mention of Christianity |
| Waterbirds | Photos | 11,788 | Waterbird or landbird | Water or land background |
| CheXpert-device | X-rays | 156,848 | Pleural effusion | Has support device |
| MultiNLI | Sentence pairs | 412,349 | Entailment | Presence of negation words |

Table 1: We study these datasets from Liu et al. (2015); Borkan et al. (2019); Sagawa et al. (2020); Irvin et al. (2019); Williams et al. (2018) respectively. For each dataset, we form a group for each combination of label $y \in \mathcal{Y}$ and spuriously-correlated attribute $a \in \mathcal{A}$, and evaluate the accuracy of selective classifiers on average and on each group. Dataset details in Appendix C.1.

**Datasets.** We consider five datasets (Table 1) on which prior work has shown that models latch onto spurious correlations, thereby performing well on average but poorly on the groups of data where the spurious correlation does not hold up. Following Sagawa et al. (2020), we define a set of labels $\mathcal{Y}$ as well as a set of attributes $\mathcal{A}$ that are spuriously correlated with the labels, and then form one group for each $(y, a) \in \mathcal{Y} \times \mathcal{A}$. For example, in the pleural effusion example from Section 1, one group would be patients with pleural effusion ($y = 1$) but no support devices ($a = 0$). Each dataset has $|\mathcal{Y}| = |\mathcal{A}| = 2$, except MultiNLI, which has $|\mathcal{Y}| = 3$. More dataset details are in Appendix C.1.

## 4 EVALUATING SELECTIVE CLASSIFICATION ON GROUPS

We start by investigating how selective classification affects group (selective) accuracies across the five datasets in Table 1. We train standard models with empirical risk minimization, i.e., to minimize average training loss, using ResNet50 for CelebA and Waterbirds; DenseNet121 for CheXpert-device; and BERT for CivilComments and MultiNLI. Details are in Appendix C. We focus on soft-max response (SR) selective classifiers, but show similar results for MC-dropout in Appendix B.1.

**Accuracy-coverage curves.** Figure 2 shows group accuracy-coverage curves for each dataset, with the average in blue, worst group in red, and other groups in gray. On all datasets, average accuracies improve as coverage decreases. However, the worst-group curves fall into three categories:

1. *Decreasing.* Strikingly, on CelebA, worst-group accuracy *decreases* with coverage: the more confident the model is on worst-group points, the more likely it is incorrect.
2. *Mixed.* On Waterbirds, CheXpert-device, and CivilComments, as coverage decreases, worst-group accuracy sometimes increases (though not by much, except at noisy, low coverages) and sometimes decreases.
3. *Slowly increasing.* On MultiNLI, as coverage decreases, worst-group accuracy consistently improves but more slowly than other groups: from full to 50% average coverage, worst-group accuracy goes from 65% to 75% while the second-to-worst group accuracy goes from 77% to 95%.

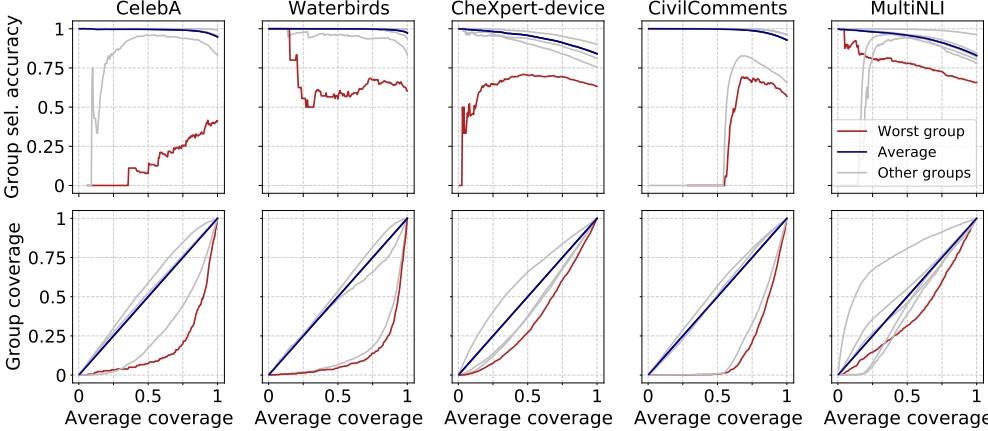

Figure 2: Accuracy (top) and coverage (bottom) for each group, as a function of the average coverage. Each average coverage corresponds to a threshold $\tau$. The red lines represent the worst group. At low coverages, accuracy estimates are noisy as only a few predictions are made.

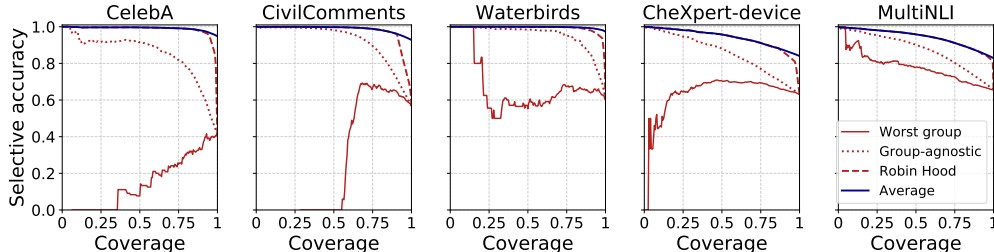

Figure 3: SR selective classifiers (solid line) substantially underperform their group-agnostic references (dotted line) on the worst group, which are in turn far behind the best-case Robin Hood references (dashed line). By construction, these share the same average accuracy-coverage curves (blue line). Similar results for MC-dropout are in Figure 7.

**Group-agnostic and Robin Hood references.** The results above show that even when selective classification is helping the worst group, it seems to help other groups more. We formalize this notion by comparing the selective classifier to a matching *group-agnostic reference* that is derived from it and that tries to abstain equally across groups. At each threshold, the group-agnostic reference makes the same numbers of correct and incorrect predictions as its corresponding selective classifier, but distributes these predictions uniformly at random across points without regard to group identities (Algorithm 1). By construction, it has an identical average accuracy-coverage curve as its corresponding selective classifier, but can differ on the group accuracies. We show in Appendix A.2 that it satisfies *equalized odds* (Hardt et al., 2016) w.r.t. which points it predicts or abstains on.

The group-agnostic reference distributes abstentions equally across groups; from the perspective of closing disparities between groups, this is the least that we might hope for. Ideally, selective classification would preferentially increase worst-group accuracy until it matches the other groups. We can capture this optimistic scenario by constructing, for a given selective classifier, a corresponding *Robin Hood reference* which, as above, also makes the same number of correct and incorrect predictions (see Algorithm 2 in Appendix A.3). However, unlike the group-agnostic reference, for the Robin Hood reference, the correct predictions are not chosen uniformly at random; instead, we prioritize picking them from the worst group, then the second worst group, etc. Likewise, we prioritize picking the incorrect predictions from the best group, then the second best group, etc. This results in worst-group accuracy rapidly increasing at the cost of the best group.

Both the group-agnostic and the Robin Hood references are not algorithms that we could implement in practice without already knowing all of the groups and labels. They act instead as references: selective classifiers that preferentially benefit the worst group would have worst-group accuracy-coverage curves that lie between the group-agnostic and Robin Hood curves. Unfortunately, Figure 3 shows that SR selective classifiers substantially underperform even their group-agnostic counterparts: they disproportionately help groups that already have higher accuracies, further exacerbating the disparities between groups. We show similar results for MC-dropout in Section B.1.

---

**Algorithm 1:** Group-agnostic reference for $(\hat{y}, \hat{c})$ at threshold $\tau$

---

**Input:** Selective classifier $(\hat{y}, \hat{c})$, threshold $\tau$, test data $\mathcal{D}$
**Output:** The sets of correct predictions $\mathcal{C}_\tau^{\text{ga}} \subseteq \mathcal{D}$ and incorrect predictions $\mathcal{I}_\tau^{\text{ga}} \subseteq \mathcal{D}$ that the
   group-agnostic reference for $(\hat{y}, \hat{c})$ makes at threshold $\tau$.

**1** Let $\mathcal{C}_\tau$ be the set of all examples that $(\hat{y}, \hat{c})$ correctly predicts at threshold $\tau$:

$$\mathcal{C}_\tau = \{(x, y, g) \in \mathcal{D} \mid \hat{y}(x) = y \text{ and } \hat{c}(x) \geq \tau\}. \tag{2}$$

   Sample a subset $\mathcal{C}_\tau^{\text{ga}}$ of size $|\mathcal{C}_\tau|$ uniformly at random from $\mathcal{C}_0$, which is the set of all
   examples that $\hat{y}$ would have predicted correctly at full coverage.
**2** Let $\mathcal{I}_\tau$ be the analogous set of incorrect predictions at $\tau$:

$$\mathcal{I}_\tau = \{(x, y, g) \in \mathcal{D} \mid \hat{y}(x) \neq y \text{ and } \hat{c}(x) \geq \tau\}. \tag{3}$$

   Sample a subset $\mathcal{I}_\tau^{\text{ga}}$ of size $|\mathcal{I}_\tau|$ uniformly at random from $\mathcal{I}_0$.
**3** Return $\mathcal{C}_\tau^{\text{ga}}$ and $\mathcal{I}_\tau^{\text{ga}}$. Since $|\mathcal{C}_\tau^{\text{ga}}| = |\mathcal{C}_\tau|$ and $|\mathcal{I}_\tau^{\text{ga}}| = |\mathcal{I}_\tau|$, the group-agnostic reference makes
   the same numbers of correct and incorrect predictions as $(\hat{y}, \hat{c})$, but in a group-agnostic way.

---

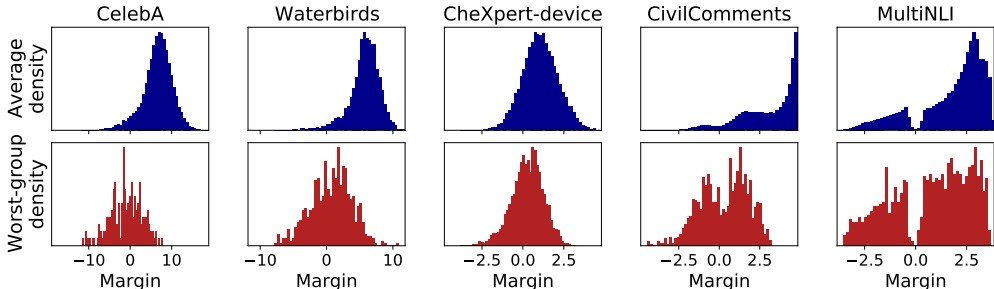

Figure 4: Margin distributions on average (top) and on the worst group (bottom). Positive (negative) margins correspond to correct (incorrect) predictions, and we abstain on points with margins closest to zero first. The worst groups have disproportionately many confident but incorrect examples.

## 5 ANALYSIS: MARGIN DISTRIBUTIONS AND ACCURACY-COVERAGE CURVES

We saw in the previous section that while selective classification typically increases average (selective) accuracy, it can either increase or decrease worst-group accuracy. We now turn to a theoretical analysis of this behavior. Specifically, we establish the conditions under which we can expect to see its two extremes: when accuracy *monotonically* increases, or decreases, as a function of coverage. While these extremes do not fully explain the empirical phenomena, e.g., why worst-group accuracy sometimes increases and then decreases, our analysis broadly captures why accuracy monotonically increases on average with decreasing coverage, but displays mixed behavior on the worst group.

Our central objects of study are the margin distributions for each group. Recall that the margin of a selective classifier $(\hat{y}, \hat{c})$ on a point $(x, y)$ is its confidence $\hat{c}(x) \geq 0$ if the prediction is correct $(\hat{y}(x) = y)$ and $-\hat{c}(x) \leq 0$ otherwise. The selective accuracies on average and on the worst-group are thus completely determined by their respective margin distributions, which we show for our datasets in Figure 4. The worst-group and average distributions are very different: the worst-group distributions are consistently shifted to the left with many confident but incorrect examples. Our approach will be to characterize what properties of a general margin distribution $F$ lead to monotonically increasing or decreasing accuracy. Then, by letting $F$ be the overall or worst-group margin distribution, we can see how the differences in these distributions lead to differences in accuracy as a function of coverage.

**Setup.** We consider distributions over margins that have a differentiable cumulative distribution function (CDF) and a density, denoted by corresponding upper- and lowercase variables (e.g., $F$ and $f$, respectively). Each margin distribution $F$ corresponds to a selective classifier over some data distribution. We denote the corresponding (selective) accuracy of the classifier at threshold $\tau$ as $A_F(\tau) = (1 - F(\tau))/(F(-\tau) + 1 - F(\tau))$. Since increasing the threshold $\tau$ monotonically decreases coverage, we focus on studying accuracy as a function of $\tau$. All proofs are in Appendix D.

### 5.1 SYMMETRIC MARGIN DISTRIBUTIONS

We begin with symmetric distributions. We introduce a generalization of log-concavity, which we call *left-log-concavity*; for symmetric distributions, left-log-concavity corresponds to monotonicity of the accuracy-coverage curve, with the direction determined by the full-coverage accuracy.

**Definition 1** (Left-log-concave distributions). *A distribution is left-log-concave if its CDF is log-concave on $(-\infty, \mu]$, where $\mu$ is the mean of the distribution.*

Left-log-concave distributions are a superset of the broad family of log-concave distributions (e.g., Gaussian, beta, uniform), which require log-concave densities (instead of CDFs) on their entire support (Boyd & Vandenberghe, 2004). Notably, they can be multimodal: a symmetric mixture of two Gaussians is left-log-concave but not generally log-concave (Lemma 1 in Appendix D).

**Proposition 1** (Left-log-concavity and monotonicity). *Let $F$ be the CDF of a symmetric distribution. If $F$ is left-log-concave, then $A_F(\tau)$ is monotonically increasing in $\tau$ if $A_F(0) \geq 1/2$ and monotonically decreasing otherwise. Conversely, if $A_{F_d}(\tau)$ is monotonically increasing for all translations $F_d$ such that $F_d(\tau) = F(\tau - d)$ for all $\tau$ and $A_{F_d}(0) \geq 1/2$, then $F$ is left-log-concave.*

Proposition 1 is consistent with the observation that selective classification tends to improve average accuracy but hurts worst-group accuracy. As an illustration, consider a margin distribution that is a symmetric mixture of two Gaussians, each corresponding to a group, and where the average accuracy is >50% at full coverage but the worst-group accuracy is <50%. As the overall and worst-group margin distributions are both left-log-concave (Lemma 1), Proposition 1 implies that worst-group accuracy will decrease monotonically with $\tau$ while the average accuracy improves monotonically.

Applied to CelebA (Figure 4), Proposition 1 is also consistent with how average accuracy improves while worst-group accuracy, which is <50% at full coverage, worsens. Finally, Proposition 1 also helps to explain why selective classification generally improves average accuracy in the literature, as average accuracies are typically high at full coverage and margin distributions often resemble Gaussians (Balasubramanian et al., 2011; Lakshminarayanan et al., 2017).

## 5.2 Skew-symmetric margin distributions

The results above apply only to symmetric margin distributions. As not all of the margin distributions in Figure 4 are symmetric, we extend our analysis to asymmetric margin distributions by building upon prior work on *skew-symmetric distributions* (Azzalini & Regoli, 2012).

**Definition 2.** *A distribution with density $f_{\alpha,\mu}$ is skew-symmetric with skew $\alpha$ and center $\mu$ if*

$$f_{\alpha,\mu}(\tau) = 2h(\tau - \mu)G(\alpha(\tau - \mu)) \tag{4}$$

*for all $\tau \in \mathbb{R}$, where $h$ is the density of a distribution is symmetric about 0, and $G$ is the CDF of a potentially different distribution that is also symmetric about 0.*

In other words, $f_{\alpha,\mu}$ is a skewed form of the symmetric density $h$, where higher $\alpha$ means more right skew, and setting $\alpha = 0$ yields a (translated) $h$. Skew-symmetric distributions are a broad family and include, e.g., skew-normal distributions, as well as all symmetric distributions. In Appendix D.3, we show some properties of skew-symmetric distributions as pertains to selective classification, e.g., margin distributions that are more right-skewed have higher accuracies (Proposition 6).

Our result is that skewing a symmetric distribution in the "same" direction preserves monotonicity: if accuracy is monotone increasing, then right skew (which increases accuracy) preserves this.

**Proposition 2** (Skew in the same direction preserves monotonicity). *Let $F_{\alpha,\mu}$ be the CDF of a skew-symmetric distribution. If accuracy of its symmetric version, $A_{F_{0,\mu}}(\tau)$, is monotonically increasing in $\tau$, then $A_{F_{\alpha,\mu}}(\tau)$ is also monotonically increasing in $\tau$ for any $\alpha > 0$. Similarly, if $A_{F_{0,\mu}}(\tau)$ is monotonically decreasing in $\tau$, then $A_{F_{\alpha,\mu}}(\tau)$ is also monotonically decreasing in $\tau$ for any $\alpha < 0$.*

## 5.3 Discussion

Proposition 1 relates the left-log-concavity of a symmetric margin distribution to the monotonicity of selective accuracy, with the direction of monotonicity (increasing or decreasing) determined by the full-coverage accuracy. Proposition 2 then states that if we have a symmetric margin distribution with monotone accuracy, skewing it in the same direction preserves monotonicity. Combining both propositions, we have that if $F_{0,\mu}$ is symmetric left-log-concave with full-coverage accuracy >50%, then the accuracy $A_{F_{\alpha,\mu}(\tau)}$ of any skewed $F_{\alpha,\mu}(\tau)$ with $\alpha > 0$ is monotone increasing in $\tau$.

Since accuracy-coverage curves are preserved under all odd, monotone transformations of margins (Lemma 8 in Appendix D), these results also generalize to odd, monotone transformations of these (skew-)symmetric distributions. As many margin distributions—in Figure 4 and in the broader literature (Balasubramanian et al., 2011; Lakshminarayanan et al., 2017)—resemble the distributions studied above (e.g., Gaussians and skewed Gaussians), we thus expect selective classification to improve average accuracy but worsen worst-group accuracy when worst-group accuracy at full coverage is low to begin with.

An open question is how to characterize the properties of margin distributions that lead to non-monotone behavior. For example, in Waterbirds and CheXpert-device, accuracies first increase and then decrease with decreasing coverage (Figure 2). These two datasets have worst-group margin distributions that have full-coverage accuracies >50% but that are *left*-skewed with skewness -0.30 and -0.33 respectively (Figure 4), so we cannot apply Proposition 2 to describe them.

## 6 Analysis: Comparison to group-agnostic reference

Even if selective classification improves worst-group accuracy, it can still exacerbate group dispari-ties, underperforming the group-agnostic reference on the worst group (Section 4). In this section, we continue our analysis and show that while it is possible to outperform the group-agnostic refer-ence, it is challenging to do so, especially when the accuracy disparity at full coverage is large.

**Setup.** Throughout this section, we decompose the margin distribution into two components $F = pF_{\text{wg}} + (1 - p)F_{\text{others}}$, where $F_{\text{wg}}$ and $F_{\text{others}}$ correspond to the margin distributions of the worst group and of all other groups combined, respectively; $p$ is the fraction of examples in the worst group; and the worst group has strictly worse accuracy at full coverage than the other groups (i.e., $A_{F_{\text{wg}}}(0) < A_{F_{\text{others}}}(0)$). Recall from Section 4 that for any selective classifier (i.e., any margin distribution), its group-agnostic reference has the same average accuracy at each threshold $\tau$ but potentially different group accuracies. We denote the worst-group accuracy of the group-agnostic reference as $\tilde{A}_{F_{\text{wg}}}(\tau)$, which can be written in terms of $F_{\text{wg}}$, $F_{\text{others}}$, and $p$ (Appendix A.2). We continue with notation from Section 5 otherwise, and all proofs are in Appendix E.

A selective classifier with margin distribution $F$ is said to outperform the group-agnostic reference on the worst group if $A_{F_{\text{wg}}}(\tau) \geq \tilde{A}_{F_{\text{wg}}}(\tau)$ for *all* $\tau \geq 0$. To establish a necessary condition for out-performing the reference, we study the neighborhood of $\tau = 0$, which corresponds to full coverage:

**Proposition 3** (Necessary condition for outperforming the group-agnostic reference). *Assume that* $1/2 < A_{F_{\text{wg}}}(0) < A_{F_{\text{others}}}(0) < 1$ *and the worst-group density* $f_{\text{wg}}(0) > 0$. *If* $\tilde{A}_{F_{\text{wg}}}(\tau) \leq A_{F_{\text{wg}}}(\tau)$ *for all* $\tau \geq 0$, *then*

$$\frac{f_{\text{others}}(0)}{f_{\text{wg}}(0)} \leq \frac{1 - A_{F_{\text{others}}}(0)}{1 - A_{F_{\text{wg}}}(0)}. \tag{5}$$

The RHS is the ratio of full-coverage errors; the larger the disparity between the worst group and the other groups at full coverage, the harder it is to satisfy this condition. In Appendix F, we simulate mixtures of Gaussians and show that this condition is rarely fulfilled.

Motivated by the empirical margin distributions, we apply Proposition 3 to the setting where $F_{\text{wg}}$ and $F_{\text{others}}$ are both log-concave and are translated and scaled versions of each other. We show that the worst group must have lower variance than the others to outperform the group-agnostic reference:

**Corollary 1** (Outperforming the group-agnostic reference requires smaller scaling for log-concave distributions). *Assume that* $1/2 < A_{F_{\text{wg}}}(0) < A_{F_{\text{others}}}(0) < 1$, $F_{\text{wg}}$ *is log-concave, and* $f_{\text{others}}(\tau) = v f_{\text{wg}}(v(\tau - \mu_{\text{others}}) + \mu_{\text{wg}})$ *for all* $\tau \in \mathbb{R}$, *where* $v$ *is a scaling factor. If* $\tilde{A}_{F_{\text{wg}}}(\tau) \leq A_{F_{\text{wg}}}(\tau)$ *for all* $\tau \geq 0$, $v < 1$.

This is consistent with the empirical margin distributions on Waterbirds: the worst group has higher variance, implying $v > 1$ as $v$ is the ratio of the worst group's standard deviation to the other group's, and it thus fails to satisfy the necessary condition for outperforming the group-agnostic reference.

A further special case is when $F_{\text{wg}}$ and $F_{\text{others}}$ are log-concave and unscaled translations of each other. Here, selective classification underperforms the group-agnostic reference at *all* thresholds $\tau$.

**Proposition 4** (Translated log-concave distributions underperform the group-agnostic reference). *Assume* $F_{\text{wg}}$ *and* $F_{\text{others}}$ *are log-concave and* $f_{\text{others}}(\tau) = f_{\text{wg}}(\tau - d)$ *for all* $\tau \in \mathbb{R}$. *Then for all* $\tau \geq 0$,

$$A_{F_{\text{wg}}}(\tau) \leq \tilde{A}_{F_{\text{wg}}}(\tau). \tag{6}$$

This helps to explain our results on CheXpert-device, where the worst-group and average margin distributions are approximately translations of each other, and selective classification significantly underperforms the group-agnostic reference at all confidence thresholds.

## 7 Selective classification on group DRO models

Our above analysis suggests that selective classification tends to exacerbate group disparities, espe-cially when the full-coverage disparities are large. This motivates a potential solution: by reducing

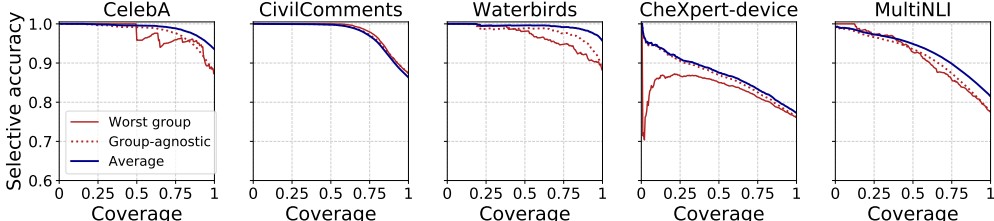

Figure 5: When applied to group DRO models, which have more similar accuracies across groups than standard models, SR selective classifiers improve average and worst-group accuracies. At low coverages, accuracy estimates are noisy as only a few predictions are made.

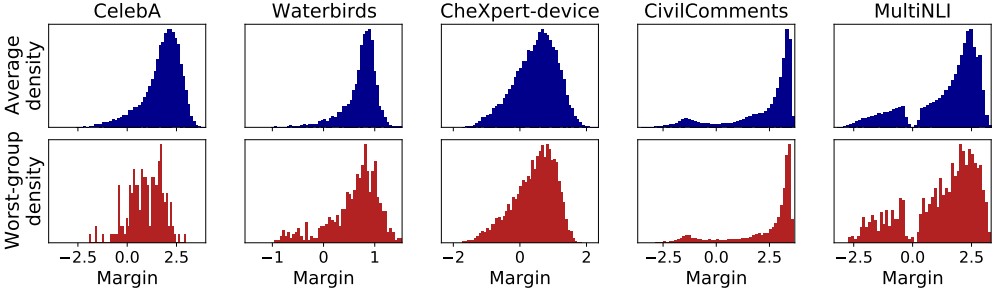

Figure 6: Density of margins for the average (top) and the worst-group (bottom) distributions over margins for the group DRO model. Positive (negative) margins correspond to correct (incorrect) predictions, and we abstain on margins closest to zero first. Unlike the selective classifiers trained with ERM, we see similar average and worst-group distributions for group DRO.

group disparities at full coverage, models are more likely to satisfy the necessary condtion for outperforming the group-agnostic reference defined in Proposition 3. In this section, we explore this approach by training models using group distributionally robust optimization (group DRO) (Sagawa et al., 2020), which minimizes the worst-group training loss $L_{\mathrm{DRO}}(\theta) = \max_{g \in \mathcal{G}} \hat{\mathbb{E}} \left[ \ell(\theta; (x, y)) \mid g \right]$. Unlike standard training, group DRO uses group annotations at training time. As with prior work, we found that group DRO models have much smaller full-coverage disparities (Figure 5). Moreover, worst-group accuracies consistently improve as coverage decreases and at a rate that is comparable to the group-agnostic reference, though small gaps remain on Waterbirds and CheXpert-device.

While our theoretical analysis motivates the above approach, the analysis ultimately depends on the margin distributions of each group, not just on their full-coverage accuracies. Although group DRO only optimizes for similar full-coverage accuracies across groups, we found that it also leads to much more similar average and worst-group margin distributions compared to ERM (Figure 6), explaining why selective classification behaves more uniformly over groups across all datasets.

Group DRO is not a silver bullet, as it relies on group annotations for training, which are not always available. Nevertheless, these results show that closing full-coverage accuracy disparities can mitigate the downstream disparities caused by selective classification.

## 8 DISCUSSION

We have shown that selective classification can magnify group disparities and should therefore be applied with caution. This is an insidious failure mode, since selective classification generally improves average accuracy and can appear to be working well if we do not look at group accuracies. However, we also found that selective classification can still work well on models that have equal full-coverage accuracies across groups. Training such models, especially without relying on too much additional information at training time, remains an important research direction. On the theoretical side, we characterized the behavior of selective classification in terms of the margin distributions; an open question is how different margin distributions arise from different data distributions, models, training procedures, and selective classification algorithms. Finally, in this paper we focused on studying selective accuracy in isolation; accounting for the cost of abstention and the equity of different coverages on different groups is an important direction for future work.

ACKNOWLEDGMENTS

We thank Emma Pierson, Jean Feng, Pranav Rajpurkar, and Tengyu Ma for helpful advice. This work was supported by NSF Award Grant no. 1804222. SS was supported by the Herbert Kunzel Stanford Graduate Fellowship and AK was supported by the Stanford Graduate Fellowship.

REPRODUCIBILITY

All code, data, and experiments are available on CodaLab at `https://worksheets.codalab.org/worksheets/0x7ceb817d53b94b0c8294a7a22643bf5e`. The code is also available on GitHub at `https://github.com/ejones313/worst-group-sc`.

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

## A SETUP

### A.1 SOFTMAX RESPONSE SELECTIVE CLASSIFIERS

In this section, we describe our implementation of softmax response (SR) selective classifiers (Geifman & El-Yaniv, 2017). Recall from Section 3 that a selective classifier is a pair $(\hat{y}, \hat{c})$, where $\hat{y} : \mathcal{X} \to \mathcal{Y}$ outputs a prediction and $\hat{c} : \mathcal{X} \to \mathbb{R}_+$ outputs the model's confidence, which is always non-negative, in that prediction. SR classifiers are defined for neural networks (for classification), which generally have a last softmax layer over the $k$ possible classes. For an input point $x$, we denote its maximum softmax probability, which corresponds to its predicted class $\hat{y}(x)$, as $\hat{p}(\hat{y}(x) \mid x)$. We defined the confidence $\hat{c}(x)$ for binary classifiers as

$$\hat{c}(x) = \frac{1}{2} \log \left( \frac{\hat{p}(\hat{y}(x) \mid x)}{1 - \hat{p}(\hat{y}(x) \mid x)} \right). \tag{7}$$

Since the maximum softmax probability $\hat{p}(\hat{y}(x) \mid x)$ is at least 0.5 for binary classification, $\hat{c}(x)$ is nonnegative for each $x$, and is thus a valid confidence. For $k > 2$ classes, however, $\hat{p}(\hat{y}(x) \mid x)$ can be less than 0.5, in which case $\hat{c}(x)$ would be negative. To ensure that confidence is always nonnegative, we define $\hat{c}(x)$ for $k$ classes to be

$$\hat{c}(x) = \frac{1}{2} \log \left( \frac{\hat{p}(\hat{y}(x) \mid x)}{1 - \hat{p}(\hat{y}(x) \mid x)} \right) + \frac{1}{2} \log(k - 1). \tag{8}$$

With $k$ classes, the maximum softmax probability $\hat{p}(\hat{y}(x) \mid x) \geq 1/k$, and therefore we can verify that $\hat{c}(x) \geq 0$ as desired. Moreover, when $k = 2$, Equation (8) reduces to our original binary confidence; we can therefore interpret the general form in Equation (8) as a normalized logit.

Note that $\hat{c}(x)$ is a monotone transformation of the maximum softmax probability $\hat{p}(\hat{y}(x) \mid x)$. Since the accuracy-coverage curve of a selective classifier only depends on the relative ranking of $\hat{c}(x)$ across points, we could have equivalently set $\hat{c}(x)$ to be $\hat{p}(\hat{y}(x) \mid x)$. However, following prior work, we choose the logit-transformed version to make the corresponding distribution of confidences easier to visualize (Balasubramanian et al., 2011; Lakshminarayanan et al., 2017).

Finally, we remark on one consequence of SR on the margin distribution for multi-class classification. Recall that we define the *margin* of an example to be $\hat{c}(x)$ on correct predictions $(\hat{y}(x) = y)$ and $-\hat{c}(x)$ otherwise, as described in Section 3. In Figure 4, we plot the margin distributions of SR selective classifiers on all five datasets. We observe that on MultiNLI, which is the only multi-class dataset (with $k = 3$), there is a gap (region of lower density) in the margin distribution around 0. We attribute this gap in part to the comparative rarity of seeing a maximum softmax probability of $\frac{1}{3}$ when $k = 3$ versus seeing $\frac{1}{2}$ when $k = 2$; in the former, all three logits must be the same, while for the latter only two logits must be the same.

### A.2 GROUP-AGNOSTIC REFERENCE

Here, we describe the group-agnostic reference described in Section 4 in more detail. We elaborate on the construction from the main text and then define the reference formally. Finally, we show that the group-agnostic reference satisfies equalized odds.

#### A.2.1 DEFINITION

We begin by recalling on the construction described in the main text for a finite test set $\mathcal{D}$. In Algorithm 1 we relied on two important sets; the set of correctly classified points at threshold $\tau$, $\mathcal{C}_\tau$:

$$\mathcal{C}_\tau = \{(x, y, g) \in \mathcal{D} \mid \hat{y}(x) = y \text{ and } \hat{c}(x) \geq \tau\}, \tag{9}$$

and similarly, the set of incorrectly classified points at threshold $\tau$, $\mathcal{I}_\tau$:

$$\mathcal{I}_\tau = \{(x, y, g) \in \mathcal{D} \mid \hat{y}(x) \neq y \text{ and } \hat{c}(x) \geq \tau\}. \tag{10}$$

To compute the accuracy of the group-agnostic reference, we sample a subset $\mathcal{C}_\tau^{\text{ga}}$ of size $|\mathcal{C}_\tau|$ uniformly at random from $\mathcal{C}_0$, and similarly a subset $\mathcal{I}_\tau^{\text{ga}}$ of size $|\mathcal{I}_\tau|$ uniformly at random from $\mathcal{I}_0$. The group-agnostic reference makes predictions when examples are in $\mathcal{C}_\tau^{\text{ga}} \cup \mathcal{I}_\tau^{\text{ga}}$ and abstains otherwise. We compute group accuracies over this set of predicted examples.

Note that the group accuracies, as defined, are randomized due to the sampling. For the remainder of our analysis, we take the expectation over this randomness to compute the group accuracies. We now generalize the above construction by considering data distributions $\mathcal{D}$. We first define the number of correctly and incorrectly classified points:

**Definition 3** (Correctly and incorrectly classified points). *Consider a selective classifier* $(\hat{y}, \hat{c})$. *For each threshold* $\tau$, *we define the fractions of points that are predicted (not abstained on), and correctly or incorrectly classified, as*

$$C(\tau) = \hat{p}(\hat{y}(x) = y \wedge \hat{c}(x) \geq \tau), \tag{11}$$

$$I(\tau) = \hat{p}(\hat{y}(x) \neq y \wedge \hat{c}(x) \geq \tau). \tag{12}$$

*We define analogous metrics for each group* $g$ *as*

$$C_g(\tau) = \hat{p}(\hat{y}(x) = y \wedge \hat{c}(x) \geq \tau \mid g), \tag{13}$$

$$I_g(\tau) = \hat{p}(\hat{y}(x) \neq y \wedge \hat{c}(x) \geq \tau \mid g). \tag{14}$$

For each threshold $\tau$, we will make predictions on a $C(\tau)/C(0)$ fraction of the $C(0)$ total (probability mass of) correctly classified points. Since each group $g$ has $C_g(0)$ correctly classified points, at threshold $\tau$, the group-agnostic reference will make predictions on $C_g(0)C(\tau)/C(0)$ correctly classified points in group $g$. We can reason similarly over the incorrectly classified points. Putting it all together, we can define the group-agnostic reference as satisfying the following:

**Definition 4** (Group-agnostic reference). *Consider a selective classifier* $(\hat{y}, \hat{c})$ *and let* $\tilde{C}, \tilde{I}, \tilde{C}_g, \tilde{I}_g$ *denote the analogous quantities to Definition 3 for its matching group-agnostic reference. For each threshold* $\tau$, *these satisfy*

$$\tilde{C}(\tau) = C(\tau) \tag{15}$$

$$\tilde{I}(\tau) = I(\tau), \tag{16}$$

*and for each threshold* $\tau$ *and group* $g$,

$$\tilde{C}_g(\tau) = C_g(0)C(\tau)/C(0) \tag{17}$$

$$\tilde{I}_g(\tau) = I_g(0)I(\tau)/I(0). \tag{18}$$

*The group-agnostic reference thus has the following accuracy on group* $g$:

$$\tilde{A}_g(\tau) = \frac{\tilde{C}_g(\tau)}{\tilde{C}_g(\tau) + \tilde{I}_g(\tau)} \tag{19}$$

$$= \frac{C_g(0)C(\tau)/C(0)}{C_g(0)C(\tau)/C(0) + I_g(0)I(\tau)/I(0)}. \tag{20}$$

### A.2.2 CONNECTION TO EQUALIZED ODDS

We now show that the group-agnostic reference satisfies equalized odds with respect to which points it predicts or abstains on. The goal of a selective classifier is to make predictions on points it would get correct (i.e., $\hat{y}(x) = y$) while abstaining on points that it would have gotten incorrect (i.e., $\hat{y}(x) \neq y$). We can view this as a meta classification problem, where a true positive is when the selective classifier decides to make a prediction on a point $x$ and gets it correct ($\hat{y}(x) = y$), and a false positive is when the selective classifier decides to make a prediction on a point $x$ and gets it incorrect ($\hat{y}(x) \neq y$). As such, we can define the *true positive rate* $R^{\mathsf{TP}}(\tau)$ and *false positive rate* $R^{\mathsf{FP}}(\tau)$ of a selective classifier:

**Definition 5.** *The true positive rate of a selective classifier at threshold* $\tau$ *is*

$$R^{\mathsf{TP}}(\tau) = \frac{C(\tau)}{C(0)}, \tag{21}$$

*and the false positive rate at threshold* $\tau$ *is*

$$R^{\mathsf{FP}}(\tau) = \frac{I(\tau)}{I(0)}. \tag{22}$$

*Analogously, the true positive and false positive rates on a group g are*

$$R_g^{TP}(\tau) = \frac{C_g(\tau)}{C_g(0)}, \tag{23}$$

$$R_g^{FP}(\tau) = \frac{I_g(\tau)}{I_g(0)}. \tag{24}$$

The group-agnostic reference satisfies equalized odds (Hardt et al., 2016) with respect to this definition:

**Proposition 5.** *The group-agnostic reference defined in Definition 4 has equal true positive and false positive rates for all groups $g \in \mathcal{G}$ and satisfies equalized odds.*

*Proof.* By construction of the group-agnostic reference (Definition 4), we have that

$$\tilde{C}_g(\tau) = C_g(0)C(\tau)/C(0) \tag{25}$$

$$= \tilde{C}_g(0)\tilde{C}(\tau)/\tilde{C}(0), \tag{26}$$

and therefore, for each group $g$, we can show that the true-positive rate of the group-agnostic reference $\tilde{R}_g^{\mathsf{TP}}$ on $g$ is equal to its average true-positive rate $\tilde{R}^{\mathsf{TP}}(\tau)$.

$$\tilde{R}_g^{\mathsf{TP}}(\tau) = \frac{\tilde{C}_g(\tau)}{\tilde{C}_g(0)} \tag{27}$$

$$= \tilde{C}(\tau)/\tilde{C}(0) \tag{28}$$

$$= \tilde{R}^{\mathsf{TP}}(\tau). \tag{29}$$

Each group thus has the same true positive rate with the group-agnostic reference. Using similar reasoning, each group also has the same false positive rate. By the definition of equalized odds, the group-agnostic reference thus satisfies equalized odds. $\square$

## A.3 ROBIN HOOD REFERENCE

In this section, we define the *Robin Hood reference*, which preferentially increases the worst-group accuracy through abstentions until it matches the other groups. Like the group-agnostic reference, we constrain the Robin Hood reference to make the same number of correct and incorrect predictions as a given selective classifier.

We formalize the definition of the Robin Hood reference in Algorithm 2. The Robin Hood reference makes predictions on the subset of examples from $\mathcal{D}$ that has the smallest discrepency between best-group and worst-group accuracies, while still matching the number of correct and incorrect predictions of a given selective classifier. Since enumerating over all possible subsets of the test data

---

**Algorithm 2:** Robin Hood reference at threshold $\tau$

**Input:** Selective classifier $(\hat{y}, \hat{c})$, threshold $\tau$, test data $\mathcal{D}$
**Output:** The set of points $\mathcal{P} \subseteq \mathcal{D}$ that the Robin Hood reference for $(\hat{y}, \hat{c})$ makes predictions on at threshold $\tau$.

1 In Algorithm 1, we defined $\mathcal{C}_\tau$ and $\mathcal{I}_\tau$ to be the sets of correct and incorrect points predicted on and abstained on at threshold $\tau$ respectively. Define $\mathcal{Q}$, the set of subsets of $\mathcal{D}$ that have the same number of correct and incorrect predictions as $(\hat{y}, \hat{c})$ at threshold $\tau$ as follows:

$$\mathcal{Q} = \{\mathcal{S} \subseteq \mathcal{D} \mid |\mathcal{P} \cap \mathcal{C}_0| = |\mathcal{C}_\tau|, |\mathcal{P} \cap \mathcal{I}_0| = |\mathcal{I}_\tau|\}. \tag{30}$$

2 Let $\mathbf{acc}_g(\mathcal{S})$ be the accuracy of $\hat{y}$ on points in $\mathcal{S}$ that belong to group $g$. We return the set $\mathcal{P}$ that minimizes the difference between the best-group and worst-group accuracies.

$$\mathcal{P} = \underset{\mathcal{S} \in \mathcal{Q}}{\operatorname{argmin}} \left( \max_{g \in \mathcal{G}} \mathbf{acc}_g(\mathcal{S}) - \min_{g \in \mathcal{G}} \mathbf{acc}_g(\mathcal{S}) \right). \tag{31}$$

---

$\mathcal{D}$ is intractable, we compute the accuracies of the Robin Hood reference iteratively. Starting from full coverage, we abstain on examples from lowest to highest confidence. Whenever we abstain on an incorrect example, we assume it comes from the current lowest accuracy group, and similarly whenever we abstain on a correctly classified example we assume it comes from the current highest accuracy group. This reduces disparities to the maximum extent possible as the threshold $\tau$ increases.

## B   SUPPLEMENTAL EXPERIMENTS

### B.1   MONTE-CARLO DROPOUT

In the main text, we observed that SR selective classifiers monotonically improve average accuracy as coverage decreases, but exacerbate accuracy disparities across groups on all five datasets. To demonstrate that these observations are not specific to SR selective classifiers, we now present our empirical results on another standard selective classification method: Monte-Carlo (MC) dropout (Gal & Ghahramani, 2016; Geifman & El-Yaniv, 2017). We find that MC-dropout selective classifiers exhibit similar empirical trends as SR selective classifiers.

**MC-dropout selective classifiers.** MC-dropout is an alternate way of assigning confidences to points. Taking a model with a dropout layer, the selective classifier first predicts $\hat{y}(x)$ simply by taking the model output without dropout. To estimate the confidence, it then samples $n$ softmax probabilities corresponding to the label $\hat{y}(x)$ over the randomness of the dropout layer. The confidence is computed as $\hat{c}(x) = 1/s$, where $s^2$ is the variance of the sampled probabilities. We implement MC-dropout by using the existing dropout layers for BERT and adding dropout to the final fully-connected layer for ResNet and DenseNet, with a dropout probability $0.1$. We present empirical results for $n = 10$; Gal & Ghahramani (2016) observed that this was sufficient to produce good confidence estimates.

**Results.** The MC-dropout selective classifiers exhibit similar trends to those that observed in Section 4, demonstrating that the observed empirical trends are not specific to SR response; even though the average accuracy improves monotonically as coverage decreases across all five datasets, the worst-group accuracy tends to decrease for CelebA, fails to increase consistently for CheXpert, Waterbirds, and CivilComments, and increases consistently but slowly for MultiNLI. Comparing SR and MC-dropout selective classifiers, we observe that the MC-dropout selective classifiers performs slightly worse. For example, we see a more prominent drop in worst-group accuracy for Waterbirds and much smaller improvements in worst-group accuracy for CheXpert.

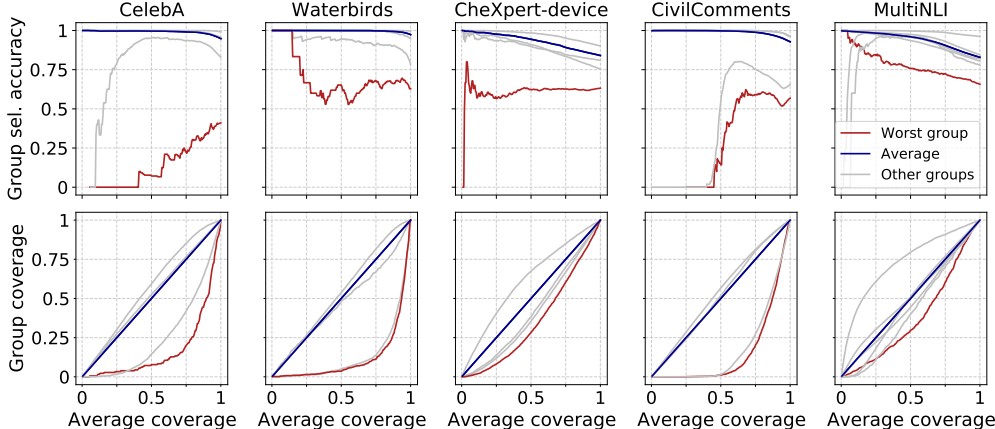

Figure 7: Selective accuracy (top) and coverage (bottom) for each group, as a function of the average coverage for the MC-dropout selective classifier. Each average coverage corresponds to a threshold $\tau$. The red lines represent the worst group.

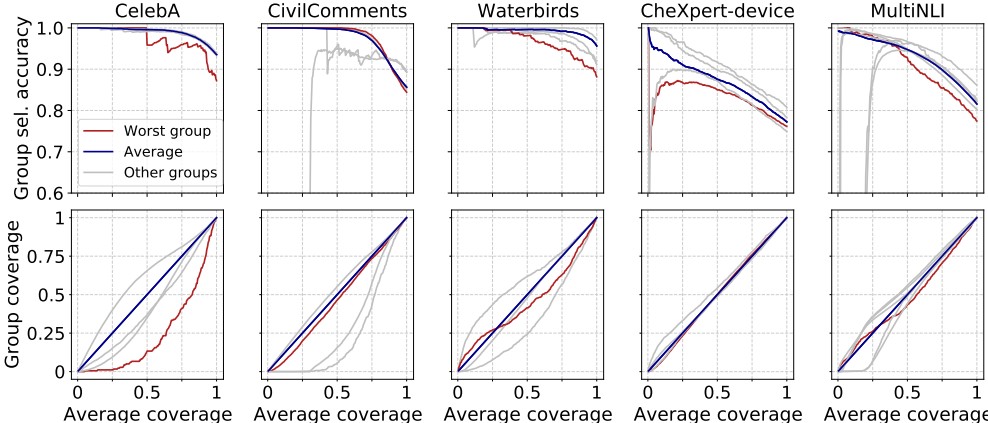

Figure 8: Selective accuracy (top) and coverage (bottom) for each group as a function of the average coverage for the softmax response selective classifier with model optimized with group DRO. Each average coverage corresponds to a specific threshold $\tau$. The red lines represent the worst group (i.e. the one with the lowest accuracy at full coverage,) and the gray lines represent the other groups.

## B.2   GROUP DRO

We showed in Section 7 that SR selective classifiers trained with the group DRO objective successfully improve worst-group selective accuracies as coverage decreases, and perform comparably to the group-agnostic reference. We now present additional empirical results for these selective classifiers.

**Accuracy-coverage curves.**   We first present the accuracy-coverage curves for all groups, along with the group coverage trends, in Figure 8. Group selective accuracies tend to improve monotonically, in stark contrast with our results on standard (ERM) selective classifiers. These trends hold generally across groups and datasets, with a few exceptions; selective accuracies drop on two groups in MultiNLI at roughly 20% average coverage, and selective accuracies improve slowly on two groups in CivilComments. Below, we look into these anomalies and offer potential explanations.

For MultiNLI, we first note that the drops are observed at very low *group* coverages—lower than 1%—at which point the accuracies are computed based on very few examples and thus are noisy. In addition, much of the anomaly can be explained by label noise; we manually inspect the 20 examples from the above two groups with the highest softmax confidences, and find that 17 of them are labeled incorrectly. The observations on CivilComments can also be potentially attributed to label noise. Removing examples with high inter-annotator disagreement (with the fraction of toxic annotations between 0.5 and 0.6) yields an accuracy-coverage curve that fits the broader empirical trends.

## C   EXPERIMENT DETAILS

### C.1   DATASETS

**CelebA.**   Models have been shown to latch onto spurious correlations between labels and demographic attributes such as race and gender (Buolamwini & Gebru, 2018; Joshi et al., 2018), and we study this on the CelebA dataset (Liu et al., 2015). Following Sagawa et al. (2020), we consider the task of classifying hair color, which is spuriously correlated with the gender. Concretely, inputs are celebrity face images, labels are hair color $\mathcal{Y} = \{\text{blond}, \text{non-blond}\}$, and spurious attributes are gender, $\mathcal{A} = \{\text{male}, \text{female}\}$, with blondness associated with being female. Of the four groups, blond males are the smallest group, with only 1,387 examples out of 162,770 training examples, and they tend to be the worst group empirically. We use the official train-val-split of the dataset.

**Waterbirds.** Object recognition models are prone to using image backgrounds as a proxy for the label (Ribeiro et al., 2016; Xiao et al., 2020). We study this on the Waterbirds dataset (Sagawa et al., 2020), constructed using images of birds from the Caltech-UCSD Birds dataset (Wah et al., 2011) placed on backgrounds from the Places dataset (Zhou et al., 2017). The task is to classify a photograph of a bird as one of $\mathcal{Y} = \{\text{waterbird}, \text{landbird}\}$, and the label is spuriously correlated with the background $\mathcal{A} = \{\text{water background}, \text{land background}\}$. Of the four groups, waterbirds on land backgrounds make up the smallest group with only 56 examples out of 4,795 training examples, and they tend to be the worst group empirically. We use the train-val-split provided by Sagawa et al. (2020), and also follow their protocol for computing average metrics; to compute average accuracies and coverages, we first compute the metrics for each group and obtain a weighted average according to group proportions in the training set, in order to account for the discrepancy in group proportions across the splits.

**CheXpert-device.** Models can latch onto spurious correlations even in high-stakes applications such as medical imaging. When models are trained to classify whether a patient has certain pathologies from chest X-rays, models have been shown to spuriously detect the presence of a support device, in particular a chest drain, instead (Oakden-Rayner et al., 2020). We study this phenomenon in a modified version of the CheXpert dataset (Irvin et al., 2019), which we call CheXpert-device. Concretely, the inputs are chest X-rays, labels are $\mathcal{Y} = \{\text{pleural effusion}, \text{no pleural effusion}\}$, and spurious attributes indicate the the presence of a support device, $\mathcal{A} = \{\text{support device}, \text{no support device}\}$. We note that chest drain is one type of support device, and is used to treat suspected pleural effusion (Porcel, 2018).

CheXpert-device is a subsampled version of the full CheXpert dataset that manifests the spurious correlation more strongly. To create CheXpert-device, we first create a new 80/10/10 train/val/test split of examples from the publicly available CheXpert train and validation sets, randomly assigning patients to splits so that all X-rays of the same patient fall in the same split. We then subsample the training set; in particular, we enforce that in 90% of examples, the label of support device matches the pleural effusion label. Of the four groups, cases of pleural effusion without a support device make up the smallest group, with 5,467 examples out of 112,100 training examples, and they tend to be the worst group empirically.

To compute the average accuracies and coverages, we weight groups according to group proportions in the training set, similarly to Waterbirds. Another complication with CheXpert is that some patients can have multiple X-rays from one visit. Following Irvin et al. (2019), we treat these images as separate training examples at training time, but output one prediction for each patient-study pair at evaluation time. Concretely, we predict pleural effusion if the model detects the condition in *any* of the X-ray images belonging to the patient-study pair, as pathologies may only appear clearly in some X-rays.

**CivilComments.** In toxicity comment detection, models have been shown to latch onto spurious correlations between the toxicity and mention of certain demographic groups (Park et al., 2018; Dixon et al., 2018). We study this in the CivilComments dataset (Borkan et al., 2019). The task is to classify the toxicity of comments on online articles with labels $\mathcal{Y} = \{\text{toxic}, \text{non-toxic}\}$. As spurious attributes, we consider whether each comment mentions a Christian identity, $\mathcal{A} = \{\text{mention of Christian identity}, \text{no mention of Christian identity}\}$; non-toxicity is associated with the mention of Christian identity, often resulting in high false negative rate on comments with such mentions. Of the four groups, toxic comments with a mention of Christian identity make up the smallest group, with only 2,446 examples out of 269,038 training examples, and they tend to be the worst group empirically. and further split the development set into a training set and a validation set by randomly splitting on articles and associating all comments with each article to either set. We use the train, validation, and test set from the WILDS benchmark (Koh et al., 2020). The training, validation, and test set all comprise comments from disjoint sets of articles. The original dataset also contains many additional examples that have toxicity annotations but not identity annotations; we do not use these in our experiments.

In the original CivilComments dataset, each comment is given a probabilistic labels for both the toxicity and the mention of a Christian identity, where a probabilistic label is the average of binary

labels across annotators. Following the associated Kaggle competition[1], we use binarized labels obtained by thresholding the probabilistic labels at 0.5.

**MultiNLI.** Lastly, we consider natural langage inference (NLI), where the task is to predict whether a hypothesis is entailed, contradicted by, or neutral to an associated premise, $\mathcal{Y} = \{\text{entailed}, \text{contradictory}, \text{neutral}\}$. NLI models have been shown to exploit annotation artifacts, for example predicting contradictory whenever negation words such as *never* or *nobody* are present (Gururangan et al., 2018). We study this on the MultiNLI dataset (Williams et al., 2018). To annotate examples' spurious attributes $\mathcal{A} = \{\text{negation words}, \text{no negation words}\}$, we consider the following negation words following Gururangan et al. (2018): "*nobody*", "*no*", "*never*", and "*nothing*". We use the splits used in Sagawa et al. (2020), whose training set includes 206,175 examples with 1,521 examples from the smallest group (entailment with negations). The worst group for standard models tends to be neutral examples with negation words.

## C.2 MODELS

We train ResNet (He et al., 2016) for CelebA and Waterbirds (images), DenseNet (Huang et al., 2017) for CheXpert (X-rays), and BERT (Devlin et al., 2019) for CivilComments and MultiNLI (text). For tasks studied in Sagawa et al. (2020) (CelebA, Waterbirds, MultiNLI), we use the hyperparameters from Sagawa et al. (2020). For others (CivilComments and CheXpert-device), we test the same number of hyperparameter sets for each of ERM and DRO, and report the best set below. Across all image and X-ray tasks, inputs are downsampled to resolution 224 x 224.

**CelebA.** To train a model on CelebA, we initialize to pretrained ResNet-50. For ERM we optimize with learning rate 1e-4, weight decay 1e-4, batch size 128, and train for 50 epochs. For DRO we use learning rate 1e-5, weight decay 1e-1, and use generalization adjustment 1 (described in Sagawa et al. (2020)). The batch size is 128, and we train for 50 epochs.

**Waterbirds.** For Waterbirds, as with CelebA, we use pretrained ResNet-50 as an initialization. For ERM we use learning rate 1e-3, weight decay 1e-4, batch size 128, and train for 300 epochs. For DRO we use learning rate 1e-5, weight decay 1, geneneralization adjustment 1, batch size 128, and train for 300 epochs.

**CheXpert-device.** For CheXpert-device, we fine-tune pretrained DenseNet-121 for three epochs. For ERM we use learning rate 1e-3, no weight decay, batch size 16, and choose the model (out of the first three epochs) with highest average accuracy (epoch 2). For DRO, we use learning rate 1e-4, weight decay 1e-1, and batch size 16, and choose the model with highest worst-group accuracy (epoch 1).

**CivilComments.** To train a model for CivilComments, we fine-tune bert-base-uncased using the implementation from Wolf et al. (2019). For both ERM and DRO we use learning rate 1e-5, weight decay 1e-2, and batch size 16. We train the ERM models and DRO models for three epochs (early stopping,) then choose the model with highest average accuracy for ERM (epoch 2), and highest worst-group accuracy for DRO (epoch 1).

**MultiNLI.** For MultiNLI we again fine-tune bert-base-uncased using the implementation from Wolf et al. (2019). For ERM, we fine-tune for three epochs with learning rate 2e-5, weight decay 0, and batch size 32. For DRO, we also use learning rate 2e-5, and weight decay 0, and batch size 32, but use generalization adjustment 1. For both ERM and DRO the model after the third epoch is best in terms of average accuracy for ERM and worst-group accuracy for DRO.

## D  PROOFS: MARGIN DISTRIBUTIONS AND ACCURACY-COVERAGE CURVES

### D.1  LEFT-LOG-CONCAVITY

Recall the definition of left-log-concave from Section 5:

**Definition 1** (Left-log-concave distributions). *A distribution is left-log-concave if its CDF is log-concave on* $(-\infty, \mu]$*, where* $\mu$ *is the mean of the distribution.*

---

[1] www.kaggle.com/c/jigsaw-unintended-bias-in-toxicity-classification/

We first prove that a symmetric mixture of Gaussians is left-log-concave, but not necessarily log-concave.

### D.1.1 SYMMETRIC MIXTURES OF GAUSSIANS ARE LEFT-LOG-CONCAVE

**Lemma 1** (Symmetric mixtures of two gaussians are left-log-concave). *Consider a symmetric mixture of Gaussians with density $f = 0.5 f_\mu + 0.5 f_{-\mu}$, where $f_\mu$ is the density of a $\mathcal{N}(\mu, \sigma^2)$ random variable and likewise $f_{-\mu}$ is the density of a $\mathcal{N}(-\mu, \sigma^2)$ random variable. Then the mixture is left-log-concave for all values of $\mu \in \mathbb{R}, \sigma > 0$, but only log-concave if $|\mu| \leq \sigma$.*

*Proof.* Without loss of generality, we can take $\sigma = 1$, since (left-)log-concavity is invariant to scaling, and also assume that $\mu$ is positive. First consider the case where $\mu \leq 1$. Then the mixture is log-concave, and therefore left-log-concave (Cule et al., 2010).

Now, consider the case where $\mu > 1$. Cule et al. (2010) show the mixture is no longer log-concave. However, we claim that it is still left-log-concave. We start by studying the gradient of $\log f$,

$$\frac{f'(x)}{f(x)} = \frac{-\exp\left(-\frac{(x-\mu)^2}{2}\right)(x-\mu) - \exp\left(-\frac{(x+\mu)^2}{2}\right)(x+\mu)}{\exp\left(-\frac{(x-\mu)^2}{2}\right) + \exp\left(-\frac{(x+\mu)^2}{2}\right)} \tag{32}$$

$$= -x + \mu \left[\frac{1 - \exp(-2x\mu)}{1 + \exp(-2x\mu)}\right]. \tag{33}$$

We claim that $\frac{f'(x)}{f(x)}$ has a local minimum at $x = -a < 0$, and as it is an odd function, a corresponding local maximum at $x = a > 0$. To show this, we first differentiate to obtain

$$\frac{d}{dx}\frac{f'(x)}{f(x)} = -1 + \frac{4\mu^2 \exp(-2x\mu)}{(1 + \exp(-2x\mu))^2}. \tag{34}$$

Setting the derivative to 0 gives us the quadratic equation

$$0 = -1 + \frac{4\mu^2 \exp(-2x\mu)}{(1 + \exp(-2x\mu))^2} \tag{35}$$

$$\Longleftrightarrow \qquad (1 + \exp(-2x\mu))^2 = 4\mu^2 \exp(-2x\mu) \tag{36}$$

$$\Longleftrightarrow \qquad [\exp(-2x\mu)]^2 + (2 - 4\mu^2)\exp(-2x\mu) + 1 = 0 \tag{37}$$

$$\Longleftrightarrow \qquad \exp(-2x\mu) = 2\mu^2 - 1 \pm 2\mu\sqrt{\mu^2 - 1}. \tag{38}$$

Since $\mu > 1$, there are two distinct roots of this quadratic, and two corresponding critical points of $f'/f$. Let $v(\mu) = 2\mu^2 - 1 + 2\mu\sqrt{\mu^2 - 1}$ be the larger root. Then $v(\mu)$ is a strictly increasing function for $\mu \geq 1$, and since $v(1) = 1$, we have that $v(\mu) > 1$ for all $\mu > 1$. Let $x = -a$ satisfy $\exp(2a\mu) = v(\mu)$. Then we have that $-a$, the smaller of the critical points of $f'/f$, is

$$-a = -\frac{\log v(\mu)}{2\mu} \tag{39}$$

$$= -\frac{\log(2\mu^2 - 1 + 2\mu\sqrt{\mu^2 - 1})}{2\mu} \tag{40}$$

$$< 0. \tag{41}$$

To show that $f'(a)/f(a)$ is a local minimum, we take the second derivative

$$\frac{d^2}{dx^2}\frac{f'(x)}{f(x)} = \frac{d}{dx}\left[-1 + \frac{4\mu^2 \exp(-2x\mu)}{(1 + \exp(-2x\mu))^2}\right] \tag{42}$$

$$= \frac{8\mu^3 \exp(-2x\mu)(\exp(-2x\mu) - 1)}{(\exp(-2x\mu) + 1)^3}, \tag{43}$$

which at $x = -a$ gives

$$\frac{d^2}{dx^2}\frac{f'(x)}{f(x)}\Big|_{x=-a} = \frac{8\mu^3 v(\mu)(v(\mu) - 1)}{(v(\mu) + 1)^3} \tag{44}$$

$$> 0, \tag{45}$$

since $v(\mu) > 1$. Since $-a$ is the only critical point of $f'/f$ that is less than 0 and it is a local minimum, $f'/f$ must be decreasing on $(-\infty, -a]$, which in turn implies that $f$, and therefore $F$, is log-concave on $(-\infty, -a]$.

It remains to show that $F$ is also log-concave on $[-a, 0]$. We make use of two facts. First, since $-a$ is a local minimum and the only critical point less than 0, we have that $\frac{f'(-a)}{f(-a)} \le \frac{f'(x)}{f(x)} \le \frac{f'(0)}{f(0)} = 0$ for all $x \in [-a, 0]$. Second, since $f(x)$ and $F(x)$ are non-negative for all $x$, $f(x)/F(x)$ is also non-negative for all $x$. Thus, for all $x \in [-a, 0]$

$$\frac{d}{dx} \frac{f(x)}{F(x)} = \frac{F(x)f'(x) - f(x)^2}{F(x)^2} \tag{46}$$

$$= \underbrace{\frac{f(x)}{F(x)}}_{\ge 0} \left( \underbrace{\frac{f'(x)}{f(x)}}_{\le 0} - \underbrace{\frac{f(x)}{F(x)}}_{\ge 0} \right) \tag{47}$$

$$\le 0, \tag{48}$$

and therefore $F$ is also log-concave on $[-a, 0]$. $\qquad\square$

**Remark 1.** *Note that if $f$ is (left-)log-concave, then $F$ is also (left-)log-concave (Bagnoli & Bergstrom, 2005). However, the reverse direction does not hold.*

### D.2 Symmetric margin distributions

In this section we prove Proposition 1. We first prove the following helpful lemma:

**Lemma 2** (Conditions for monotonicity of selective accuracy). *$A_F(\tau)$ is monotone increasing in $\tau$ if and only if*

$$\frac{f(-\tau)}{F(-\tau)} \ge \frac{f(\tau)}{1 - F(\tau)} \tag{49}$$

*for all $\tau \ge 0$. Conversely, $A_F(\tau)$ is monotone decreasing in $\tau$ if and only if the above inequality is flipped for all $\tau \ge 0$.*

*Proof.* $A_F(\tau)$ is monotone increasing in $\tau$ if and only if $\frac{dA_F}{d\tau} \ge 0$ for all $\tau \ge 0$. We obtain $\frac{dA_F}{d\tau}$ by differentiating $A_F$ and simplifying:

$$\frac{dA_F}{d\tau} = \frac{d}{d\tau} \left( \frac{1 - F(\tau)}{1 - F(\tau) + F(-\tau)} \right) \tag{50}$$

$$= \frac{\left[1 - F(\tau) + F(-\tau)\right]\left[-f(\tau)\right] - \left[1 - F(\tau)\right]\left[-f(\tau) - f(-\tau)\right]}{(1 - F(\tau) + F(-\tau))^2} \tag{51}$$

$$= \frac{-f(\tau) + f(\tau)F(\tau) - f(\tau)F(-\tau) + f(\tau) + f(-\tau) - f(\tau)F(\tau) - f(-\tau)F(\tau)}{(1 - F(\tau) + F(-\tau))^2}$$

$$= \frac{f(-\tau) - f(\tau)F(-\tau) - f(-\tau)F(\tau)}{(1 - F(\tau) + F(-\tau))^2} \tag{52}$$

$$= \frac{f(-\tau)[1 - F(\tau)] - f(\tau)F(-\tau)}{(1 - F(\tau) + F(-\tau))^2}. \tag{53}$$

Since the denominator is always positive, we have that $\frac{dA_F}{d\tau} \ge 0$ if and only if the numerator $f(-\tau)[1 - F(\tau)] - f(\tau)F(-\tau) \ge 0$, which in turn is equivalent to

$$\frac{f(-\tau)}{F(-\tau)} \ge \frac{f(\tau)}{1 - F(\tau)}, \tag{54}$$

as desired. The case for monotone decreasing $A_F(\tau)$ is analogous. $\qquad\square$

In the next two lemmas, we prove the necessary and sufficient conditions for Proposition 1 respectively.

**Lemma 3** (Left-log-concavity and symmetry imply monotonicity.). *Let $f$ be symmetric about $\mu$ and let $F$ be left-log-concave. If $A_F(0) \geq 0.5$, then $A_F(\tau)$ is monotone increasing. Conversely, if $A_F(0) \leq 0.5$, then $A_F(\tau)$ is monotone decreasing.*

*Proof.* Consider the case where $A_F(0) \geq 0.5$; the case where $A_F(0) \leq 0.5$ is analogous. From Lemma 2 and the symmetry of $f$, we have that $A_F(\tau)$ is monotone increasing if (and only if)

$$\frac{f(-\tau)}{F(-\tau)} \geq \frac{f(\tau)}{1 - F(\tau)} = \frac{f(2\mu - \tau)}{F(2\mu - \tau)} \tag{55}$$

holds for all $\tau \geq 0$.

To show that this inequality holds for all $\tau \geq 0$, we first note that since $A_F(0) \geq 0.5$, we have that $F(0) \leq 0.5$, which together with the symmetry of $F$ implies that $\mu \geq 0$. Thus, $-\tau \leq 2\mu - \tau$ for all $\tau \geq 0$.

Now, if $\tau \geq \mu$, then $2\mu - \tau \leq \mu$, so the desired inequality (55) follows from the log-concavity of $F$ on $(-\infty, \mu]$ (Remark 1 of Bagnoli & Bergstrom (2005).)

If instead $0 \leq \tau \leq \mu$, we have

$$\frac{f(-\tau)}{F(-\tau)} \geq \frac{f(\tau)}{F(\tau)} \qquad \text{[by log-concavity of $F$ on $(-\infty, \mu]$]} \tag{56}$$

$$\geq \frac{f(\tau)}{1 - F(\tau)} \qquad \text{[since $F(\tau) \leq 0.5$]} \tag{57}$$

and thus (55) also holds. $\qquad\square$

**Lemma 4** (Monotonicity and symmetry imply left-log-concavity.). *Let $f$ be symmetric with mean 0, and let $f_\mu$ be a translated version of $f$ that has mean $\mu$. If $A_{f_\mu}$ is monotone increasing for all $\mu \geq 0$, then $f$ is left-log-concave.*

*Proof.* First consider $\mu \geq 0$. Since $A_{f_\mu}$ is monotone increasing for all $\mu \geq 0$, from Lemma 2 and the symmetry of $f_\mu$,

$$\frac{f_\mu(-\tau)}{F_\mu(-\tau)} \geq \frac{f_\mu(\tau)}{1 - F_\mu(\tau)} = \frac{f_\mu(2\mu - \tau)}{F_\mu(2\mu - \tau)} \tag{58}$$

must hold for all $\tau \geq 0$. Since $f_\mu(x) = f(x - \mu)$ by construction, we can equivalently write

$$\frac{f(-\mu - \tau)}{F(-\mu - \tau)} \geq \frac{f(\mu - \tau)}{F(\mu - \tau)}, \tag{59}$$

which holds for all $\tau \geq 0$ as well, and by assumption, for all $\mu \geq 0$ as well. By letting $a = -\mu - \tau$ and $b = \mu - \tau$, we can equivalently write the above inequality as

$$\frac{f(a)}{F(a)} \geq \frac{f(b)}{F(b)} \tag{60}$$

for all $a, b \in \mathbb{R}$ where $a \leq b, a + b \leq 0$.

By the definition of log-concavity, this means that $F$ is log-concave on $(-\infty, 0]$. $\qquad\square$

### D.2.1   LEFT-LOG-CONCAVITY AND MONOTONICITY.

**Proposition 1** (Left-log-concavity and monotonicity). *Let $F$ be the CDF of a symmetric distribution. If $F$ is left-log-concave, then $A_F(\tau)$ is monotonically increasing in $\tau$ if $A_F(0) \geq 1/2$ and monotonically decreasing otherwise. Conversely, if $A_{F_d}(\tau)$ is monotonically increasing for all translations $F_d$ such that $F_d(\tau) = F(\tau - d)$ for all $\tau$ and $A_{F_d}(0) \geq 1/2$, then $F$ is left-log-concave.*

*Proof.* If $F$ is left-log-concave, by Lemma 3 $A_F(\tau)$ is monotonically increasing in $\tau$ if $A_{F_\mu}(0) \geq \frac{1}{2}$ and monotonically decreasing otherwise. Conversely, if $A_F$ is monotonically increasing for all translations $F_d$ such that $F_d(\tau) = F(\tau - d)$ for all $\tau$ by Lemma 4, $F$ is log-concave, completing the proof. $\qquad\square$

### D.3 SKEW-SYMMETRIC MARGIN DISTRIBUTIONS

We now turn to how skew affects selective classification. Recall in Section 5 we defined skew-symmetric distributions as follows:

**Definition 2.** *A distribution with density $f_{\alpha,\mu}$ is skew-symmetric with skew $\alpha$ and center $\mu$ if*

$$f_{\alpha,\mu}(\tau) = 2h(\tau - \mu)G(\alpha(\tau - \mu)) \tag{4}$$

*for all $\tau \in \mathbb{R}$, where $h$ is the density of a distribution is symmetric about 0, and $G$ is the CDF of a potentially different distribution that is also symmetric about 0.*

When $\alpha = 0$, $f = h$ and there is no skew. Increasing $\alpha$ increases rightward skew, and decreasing $\alpha$ increases leftward skew. Note that in general, the mean of $f$ depends on $\alpha$ as well. We will also consider the translated distribution:

$$f_{\alpha,\mu}(x) = f_\alpha(x - \mu). \tag{61}$$

The CDF of $f_{\alpha,\mu}$ is $F_{\alpha,\mu}$ and the CDF of $f_\alpha$ is $F_\alpha$. We will use the following properties of these distributions:

**Lemma 5** (Skew-symmetry about $\mu$). *These properties hold when we flip the skew from $\alpha$ to $-\alpha$:*

$$f_{\alpha,\mu}(x) = f_{-\alpha,\mu}(2\mu - x) \tag{62}$$
$$F_{\alpha,\mu}(x) = 1 - F_{-\alpha,\mu}(2\mu - x) \tag{63}$$

*Proof.*

$$f_{\alpha,\mu}(x) = 2h(x - \mu)G(\alpha(x - \mu)) \tag{64}$$
$$= 2h(\mu - x)[G((-\alpha)(\mu - x))] \tag{65}$$
$$= f_{-\alpha,\mu}(2\mu - x). \tag{66}$$

$$F_{\alpha,\mu}(x) = \int_{-\infty}^{x} f_{\alpha,\mu}(t)dt \tag{67}$$

$$= \int_{-\infty}^{x} f_{-\alpha,\mu}(2c - t)dt \tag{68}$$

$$= \int_{2c-x}^{\infty} f_{-\alpha,\mu}(t)dt \tag{69}$$

$$= 1 - F_{-\alpha,\mu}(2c - x). \tag{70}$$

$\square$

**Lemma 6** (Stochastic ordering with $\alpha$ [Proposition 4 from Azzalini & Regoli (2012)]). *Let $\alpha_1 \leq \alpha_2$. Then, for any $\mu \in \mathbb{R}$, $F_{\alpha_1,\mu} \geq F_{\alpha_2,\mu}$.*

*Proof.* Since the ordering is invariant to translation, without loss of generality we can take $\mu = 0$. First consider $x \leq 0$. Then

$$F_{\alpha_1}(x) = \int_{-\infty}^{x} 2h(t)G(\alpha_1 t)dt \tag{71}$$

$$\geq \int_{-\infty}^{x} 2h(t)G(\alpha_2 t)dt \tag{72}$$

$$= F_{\alpha_2}(x). \tag{73}$$

Now consider $x \geq 0$. We have

$$F_{\alpha_1}(x) = \int_{-\infty}^{x} 2h(t)G(\alpha_1 t)dt \tag{74}$$

$$= \int_{-\infty}^{x} 2h(t)(1 - G(-\alpha_1 t))dt \qquad \text{[by symmetry of G]} \tag{75}$$

$$= 2H(x) - \int_{-\infty}^{x} 2h(t)G(-\alpha_1 t)dt \tag{76}$$

$$= 2H(x) - \int_{-\infty}^{x} 2h(-t)G(-\alpha_1 t)dt \qquad \text{[by symmetry of h]} \tag{77}$$

$$= 2H(x) - \int_{-x}^{\infty} 2h(t)G(\alpha_1 t)dt \qquad \text{[change of variables]} \tag{78}$$

$$= 2H(x) - 1 + F_{\alpha_1}(-x), \tag{79}$$

which reduces to the case where $x \leq 0$. $\qquad\square$

**Lemma 7** (Log gradient ordering by skew). *For all $\alpha \geq 0$, $\tau \geq 0$, and $\mu \in \mathbb{R}$,*

$$\frac{f_{\alpha,\mu}(\tau)}{F_{\alpha,\mu}(\tau)} \geq \frac{f_{0,\mu}(\tau)}{F_{0,\mu}(\tau)} \geq \frac{f_{-\alpha,\mu}(\tau)}{F_{-\alpha,\mu}(\tau)}. \tag{80}$$

*Proof.* Since the ordering is invariant to translation, without loss of generality we can take $\mu = 0$. We have:

$$\frac{f_\alpha(\tau)}{F_\alpha(\tau)} = \frac{h(\tau)G(\alpha\tau)}{\int_{-\infty}^{\tau} h(t)G(\alpha t)dt} \tag{81}$$

$$= \frac{h(\tau)}{\int_{-\infty}^{\tau} h(t)\frac{G(\alpha t)}{G(\alpha \tau)}dt} \tag{82}$$

$$\geq \frac{h(\tau)}{\int_{-\infty}^{\tau} h(t)dt} = \frac{f_0(\tau)}{f_0(\tau)} \tag{83}$$

$$\geq \frac{h(\tau)}{\int_{-\infty}^{\tau} h(t)\frac{G(-\alpha t)}{G(-\alpha \tau)}dt} \tag{84}$$

$$= \frac{h(\tau)G(-\alpha\tau)}{\int_{-\infty}^{\tau} h(t)G(-\alpha t)dt} \tag{85}$$

$$= \frac{f_{-\alpha}(\tau)}{F_{-\alpha}(\tau)}. \tag{86}$$

To get the inequalities, note that when $\alpha \geq 0$ and $t \leq \tau$, we have $\alpha \geq 0$, $G(\alpha t)/G(\alpha\tau) \leq 1$ since $G$ is an increasing function. Similarly, $G(-\alpha t)/G(-\alpha\tau) \geq 1$. Since $h$ is non-negative, the inequalities then follow from the monotonicity of the integral. $\qquad\square$

We now prove the main results of this section.

### D.3.1 ACCURACY IS MONOTONE WITH SKEW

**Proposition 6** (Accuracy is monotone with skew). *Let $F_{\alpha,\mu}$ be the CDF of a skew-symmetric distribution. For all $\tau \geq 0$ and $\mu \in \mathbb{R}$, $A_{F_{\alpha,\mu}}(\tau)$ is monotonically increasing in $\alpha$.*

*Proof.* We use the skew-symmetry of $f_{\alpha,\mu}$ to write the selective accuracy as

$$A_{f_{\alpha,\mu}}(\tau) = \frac{1}{1 + \frac{F_{\alpha,\mu}(-\tau)}{1 - F_{\alpha,\mu}(\tau)}} \tag{87}$$

$$= \frac{1}{1 + \frac{F_{\alpha,\mu}(-\tau)}{F_{\alpha,\mu}(2\mu-\tau)}}. \tag{88}$$

We see that $A_{f_{\alpha,\mu}}(\tau)$ is a monotone decreasing function of $\frac{F_{\alpha,\mu}(-\tau)}{F_{-\alpha,\mu}(2\mu-\tau)}$. From Lemma 6, the numerator decreases with increasing $\alpha$ while the denominator increases. Thus, this fraction decreases, which in turn implies that $A_{f_{\alpha,\mu}}(\tau)$ is monotone increasing with $\alpha$ as desired. $\square$

### D.3.2 SKEW IN THE SAME DIRECTION PRESERVES MONOTONICITY

**Proposition 2** (Skew in the same direction preserves monotonicity). *Let $F_{\alpha,\mu}$ be the CDF of a skew-symmetric distribution. If accuracy of its symmetric version, $A_{F_{0,\mu}}(\tau)$, is monotonically increasing in $\tau$, then $A_{F_{\alpha,\mu}}(\tau)$ is also monotonically increasing in $\tau$ for any $\alpha > 0$. Similarly, if $A_{F_{0,\mu}}(\tau)$ is monotonically decreasing in $\tau$, then $A_{F_{\alpha,\mu}}(\tau)$ is also monotonically decreasing in $\tau$ for any $\alpha < 0$.*

*Proof.* The idea is to use Lemma 7 to reduce the statement to the case where $\alpha = 0$, so that we can apply monotonicity. First consider the case where $A_{F_{0,\mu}}(\tau)$ is monotonically increasing in $\tau$, and $\alpha > 0$. We have

$$\frac{f_{\alpha,\mu}(-\tau)}{F_{\alpha,\mu}(-\tau)} \geq \frac{f_{0,\mu}(-\tau)}{F_{0,\mu}(-\tau)} \qquad \text{[Lemma 7]} \qquad (89)$$

$$= \frac{h_\mu(-\tau)}{H_\mu(-\tau)} \qquad \text{[definition of } f] \qquad (90)$$

$$\geq \frac{h_\mu(\tau)}{1 - H_\mu(\tau)} \qquad \text{[from monotonicity of } A \text{ (Lemma 2)]} \qquad (91)$$

$$= \frac{h_\mu(2\mu - \tau)}{H_\mu(2\mu - \tau)} \qquad \text{[symmetry of } h] \qquad (92)$$

$$= \frac{f_{0,\mu}(2\mu - \tau)}{F_{0,\mu}(2\mu - \tau)} \qquad \text{[definition of } f] \qquad (93)$$

$$\geq \frac{f_{-\alpha,\mu}(2\mu - \tau)}{F_{-\alpha,\mu}(2\mu - \tau)} \qquad \text{[Lemma 7]} \qquad (94)$$

$$\geq \frac{f_{\alpha,\mu}(\tau)}{1 - F_{\alpha,\mu}(\tau)}. \qquad \text{[Lemma 5]} \qquad (95)$$

Applying Lemma 2 completes the proof. The case where $\alpha < 0$ is analogous. $\square$

### D.4 MONOTONE ODD TRANSFORMATIONS PRESERVE ACCURACY-COVERAGE CURVES.

We now show that our results are unchanged by strictly monotone and odd transformations to the density.

**Lemma 8** (Odd and strictly monotonically increasing transformations preserve selective accuracy). *Let $X$ be a real-valued random variable with CDF $F_X$, $T$ be a strictly monotonically increasing function, and define random variable $Y = T(X)$ with CDF $F_Y$. Then, $A_{F_X}(\tau) = A_{F_Y}(T(\tau))$ for each $\tau$.*

*Proof.* For each $\tau$, since $T$ is a strictly monotonically increasing function, we apply the change of variables formula for transformations of univariate random variables to get the following:

$$F_X(\tau) = F_Y(T(\tau)) \qquad (96)$$
$$F_X(-\tau) = F_Y(T(-\tau)) = F_Y(-T(\tau)). \qquad \text{[}T \text{ is odd]} \qquad (97)$$

We now solve for selective accuracy:

$$A_{F_X}(\tau) = \frac{1 - F_X(\tau)}{1 - F_X(\tau) + F_X(-\tau)} \qquad (98)$$

$$= \frac{1 - F_Y(T(\tau))}{1 - F_Y(T(\tau)) + F_Y(-T(\tau))} \qquad (99)$$

$$= A_{F_Y}(T(\tau)). \qquad (100)$$

$\square$

With Lemma 8, our results extend all random variables $T(X)$ where $X$ corresponds to a left-log-concave distribution and $T$ is odd and strictly monotonically increasing. In particular, $X$ and $T(X)$ have the same accuracy-coverage curves.

## E    PROOFS: COMPARISON TO THE GROUP-AGNOSTIC REFERENCE

In this section, we present the proofs from Section 6, which outline conditions under which selective classifiers outperform the group-agnostic reference.

### E.1    DEFINITIONS

We first define certain metrics on selective classifiers and their group-agnostic reference in terms of their margin distributions.

**Definition 6.** *Consider a margin distribution with CDF*

$$F = \sum_{g \in \mathcal{G}} p_g F_g, \tag{101}$$

*where $p_g$ is the mixture weight and $F_g$ is the CDF for group $g$. We write the fraction of examples that are correct/incorrect and predicted on at threshold $\tau$ from group $g$ and on average as follows:*

$$C_g(\tau) = 1 - F_g(\tau) \tag{102}$$
$$I_g(\tau) = F_g(-\tau) \tag{103}$$
$$C(\tau) = \sum_g p_g C_g(\tau) \tag{104}$$
$$I(\tau) = \sum_g p_g I_g(\tau) \tag{105}$$

*We write the true-positive rate $R^{TP}(\tau)$ and false-positive rate $R^{FP}(\tau)$ for a given threshold $\tau$ as*

$$R^{TP}(\tau) = \frac{C(\tau)}{C(0)}, \tag{106}$$

$$R^{FP}(\tau) = \frac{I(\tau)}{I(0)}. \tag{107}$$

*Finally, we write the accuracy of the group-agnostic reference on group $g$ as*

$$\tilde{A}_{F_g}(\tau) = \frac{A_{F_g}(0) R^{TP}(\tau)}{A_{F_g}(0) R^{TP}(\tau) + (1 - A_{F_g}(0)) R^{FP}(\tau)} \tag{108}$$

$$= \frac{C_g(0) C(\tau)/C(0)}{C_g(0) C(\tau)/C(0) + I_g(0) I(\tau)/I(0)} \tag{109}$$

For convenience in our proofs below, we also define the fraction of each group $g$ in correctly and incorrectly classified predictions at each threshold $\tau$.

**Definition 7.** *We define the fraction of group $g$ out of correctly and incorrectly classified predictions at threshold $\tau$, $\mathbf{CF}_g(\tau)$ and $\mathbf{IF}_g(\tau)$ respectively, as*

$$\mathbf{CF}_g(\tau) = \frac{p_g C_g(\tau)}{C(\tau)} \tag{110}$$

$$\mathbf{IF}_g(\tau) = \frac{p_g I_g(\tau)}{I(\tau)}. \tag{111}$$

### E.2    GENERAL NECESSARY CONDITIONS TO OUTPERFORM THE GROUP-AGNOSTIC REFERENCE

We now present the proofs for Proposition 3 and Corollary 1, starting with the supporting lemmas.

We first write the accuracy of the group-agnostic reference in terms of $\mathbf{IF}_{wg}(0)$ and $\mathbf{CF}_{wg}(0)$.

**Lemma 9.**

$$\tilde{A}_{F_{\mathsf{wg}}}(\tau) = \frac{1}{1 + \frac{\mathbf{IF}_{\mathsf{wg}}(0)}{\mathbf{CF}_{\mathsf{wg}}(0)} \cdot \frac{I(\tau)}{C(\tau)}}. \tag{112}$$

*Proof.* We have:

$$\tilde{A}_{F_{\mathsf{wg}}}(\tau) = \frac{A_{F_{\mathsf{wg}}}(0)R^{\mathsf{TP}}(\tau)}{A_{F_{\mathsf{wg}}}(0)R^{\mathsf{TP}}(\tau) + (1 - A_{F_{\mathsf{wg}}}(0))R^{\mathsf{FP}}(\tau)} \tag{113}$$

$$= \frac{pC_{\mathsf{wg}}(0)R^{\mathsf{TP}}(\tau)}{pC_{\mathsf{wg}}(0)R^{\mathsf{TP}}(\tau) + pI_{\mathsf{wg}}(0)R^{\mathsf{FP}}(\tau)} \tag{114}$$

$$= \frac{pC_{\mathsf{wg}}(0)\frac{C(\tau)}{C(0)}}{pC_{\mathsf{wg}}(0)\frac{C(\tau)}{C(0)} + pI_{\mathsf{wg}}(0)\frac{I(\tau)}{I(0)}} \tag{115}$$

$$= \frac{\mathbf{CF}_{\mathsf{wg}}(0)C(\tau)}{\mathbf{CF}_{\mathsf{wg}}(0)C(\tau) + \mathbf{IF}_{\mathsf{wg}}(0)I(\tau)} \tag{116}$$

$$= \frac{1}{1 + \frac{\mathbf{IF}_{\mathsf{wg}}(0)}{\mathbf{CF}_{\mathsf{wg}}(0)} \cdot \frac{I(\tau)}{C(\tau)}}. \tag{117}$$

$\square$

**Lemma 10** (Bounding the derivative of $1/\mathbf{CF}_{\mathsf{wg}}(\tau)$ at $\tau = 0$). *If $A_{F_{\mathsf{others}}}(0) \geq A_{F_{\mathsf{wg}}}(0)$,*

$$\frac{d}{d\tau}\left(\frac{1}{\mathbf{CF}_{\mathsf{wg}}(\tau)}\right)\Big|_{\tau=0} \geq \left(\frac{1-p}{p}\right)\left(\frac{F_{\mathsf{wg}}(0)}{1 - F_{\mathsf{wg}}(0)}\right)\left(\frac{f_{\mathsf{wg}}(0)F_{\mathsf{others}}(0) - f_{\mathsf{others}}(0)F_{\mathsf{wg}}(0)}{F_{\mathsf{wg}}(0)^2}\right) \tag{118}$$

*Proof.*

$$\frac{d}{d\tau}\left(\frac{1}{\mathbf{CF}_{\mathsf{wg}}(\tau)}\right)\Big|_{\tau=0} \tag{119}$$

$$= \frac{d}{d\tau}\left(\frac{pC_{\mathsf{wg}}(\tau) + (1-p)C_{\mathsf{others}}(\tau)}{pC_{\mathsf{wg}}(\tau)}\right)\Big|_{\tau=0} \tag{120}$$

$$= \frac{d}{d\tau}\left(1 + \left(\frac{1-p}{p}\right)\left(\frac{1 - F_{\mathsf{others}}(\tau)}{1 - F_{\mathsf{wg}}(\tau)}\right)\right)\Big|_{\tau=0} \tag{121}$$

$$= \frac{1-p}{p}\frac{d}{d\tau}\left(\frac{1 - F_{\mathsf{others}}(\tau)}{1 - F_{\mathsf{wg}}(\tau)}\right)\Big|_{\tau=0} \tag{122}$$

$$= \frac{1-p}{p}\left(\frac{f_{\mathsf{wg}}(\tau)(1 - F_{\mathsf{others}}(\tau)) - f_{\mathsf{others}}(\tau)(1 - F_{\mathsf{wg}}(\tau))}{(1 - F_{\mathsf{wg}}(\tau))^2}\right)\Big|_{\tau=0} \tag{123}$$

$$= \frac{1-p}{p}\frac{1}{1 - F_{\mathsf{wg}}(\tau)}\left(f_{\mathsf{wg}}(\tau)\left(\frac{1 - F_{\mathsf{others}}(\tau)}{1 - F_{\mathsf{wg}}(\tau)}\right) - f_{\mathsf{others}}(\tau)\right)\Big|_{\tau=0} \tag{124}$$

$$= \frac{1-p}{p}\frac{1}{1 - F_{\mathsf{wg}}(0)}\left(f_{\mathsf{wg}}(0)\left(\frac{1 - F_{\mathsf{others}}(0)}{1 - F_{\mathsf{wg}}(0)}\right) - f_{\mathsf{others}}(0)\right) \tag{125}$$

$$\geq \frac{1-p}{p}\frac{1}{1 - F_{\mathsf{wg}}(0)}\left(f_{\mathsf{wg}}(0)\frac{F_{\mathsf{others}}(0)}{F_{\mathsf{wg}}(0)} - f_{\mathsf{others}}(0)\right) \qquad [0 < F_{\mathsf{others}}(0) \leq F_{\mathsf{wg}}(0) < 1] \tag{126}$$

$$= \frac{1-p}{p}\frac{1}{1 - F_{\mathsf{wg}}(0)}\frac{f_{\mathsf{wg}}(0)F_{\mathsf{others}}(0) - f_{\mathsf{others}}(0)F_{\mathsf{wg}}(0)}{F_{\mathsf{wg}}(0)} \tag{127}$$

$$= \frac{1-p}{p}\frac{F_{\mathsf{wg}}(0)}{1 - F_{\mathsf{wg}}(0)}\frac{f_{\mathsf{wg}}(0)F_{\mathsf{others}}(0) - f_{\mathsf{others}}(0)F_{\mathsf{wg}}(0)}{F_{\mathsf{wg}}(0)^2} \tag{128}$$

$\square$

**Lemma 11.** *If $A_{F_{\text{others}}}(0) \geq A_{F_{\text{wg}}}(0) > 0.5$, then*

$$\frac{d}{d\tau} \frac{\mathbf{IF}_{\text{wg}}(\tau)}{\mathbf{CF}_{\text{wg}}(\tau)}\bigg|_{\tau=0} \geq C(f_{\text{others}}(0)F_{\text{wg}}(0) - f_{\text{wg}}(0)F_{\text{others}}(0)) \tag{129}$$

*for some positive constant $C$.*

*Proof.*

$$\frac{d}{d\tau} \frac{\mathbf{IF}_{\text{wg}}(\tau)}{\mathbf{CF}_{\text{wg}}(\tau)}\bigg|_{\tau=0} \tag{130}$$

$$= \left(\frac{d}{d\tau}\mathbf{IF}_{\text{wg}}(\tau)\bigg|_{\tau=0}\right) \frac{1}{\mathbf{CF}_{\text{wg}}(0)} + \mathbf{IF}_{\text{wg}}(0)\left(\frac{d}{d\tau}\frac{1}{\mathbf{CF}_{\text{wg}}(\tau)}\bigg|_{\tau=0}\right) \tag{131}$$

$$\geq \left(\frac{d}{d\tau}\mathbf{IF}_{\text{wg}}(\tau)\bigg|_{\tau=0}\right) \frac{1}{\mathbf{CF}_{\text{wg}}(0)} \tag{132}$$

$$+ \mathbf{IF}_{\text{wg}}(0)\left(\frac{1-p}{p}\right)\left(\frac{F_{\text{wg}}(0)}{1 - F_{\text{wg}}(0)}\right)\left(\frac{f_{\text{wg}}(0)F_{\text{others}}(0) - f_{\text{others}}(0)F_{\text{wg}}(0)}{F_{\text{wg}}(0)^2}\right) \tag{133}$$

$$= \left(\frac{1}{1 + \frac{1-p}{p}\frac{F_{\text{others}}(0)}{F_{\text{wg}}(0)}}\right)^2 \left(\frac{1-p}{p}\right)\left(\frac{f_{\text{others}}(0)F_{\text{wg}}(0) - f_{\text{wg}}(0)F_{\text{others}}(0)}{F_{\text{wg}}(0)^2}\right) \tag{134}$$

$$* \left(1 + \left(\frac{1-p}{p}\right)\left(\frac{1 - F_{\text{others}}(0)}{1 - F_{\text{wg}}(0)}\right)\right) \tag{135}$$

$$+ \left(\frac{1}{1 + \frac{1-p}{p}\frac{F_{\text{others}}(0)}{F_{\text{wg}}(0)}}\right)\left(\frac{1-p}{p}\right)\left(\frac{F_{\text{wg}}(0)}{1 - F_{\text{wg}}(0)}\right)\left(\frac{f_{\text{wg}}(0)F_{\text{others}}(0) - f_{\text{others}}(0)F_{\text{wg}}(0)}{F_{\text{wg}}(0)^2}\right)$$

$$= \underbrace{\left(\frac{1}{1 + \frac{1-p}{p}\frac{F_{\text{others}}(0)}{F_{\text{wg}}(0)}}\right)}_{>0} \underbrace{\left(\frac{1-p}{p}\right)}_{>0} (f_{\text{others}}(0)F_{\text{wg}}(0) - f_{\text{wg}}(0)F_{\text{others}}(0))\underbrace{\left(\frac{1}{F_{\text{wg}}(0)^2}\right)}_{>0} \tag{136}$$

$$* \left(\underbrace{\frac{1 + \frac{1-p}{p}\frac{1 - F_{\text{others}}(0)}{1 - F_{\text{wg}}(0)}}{1 + \frac{1-p}{p}\frac{F_{\text{others}}(0)}{F_{\text{wg}}(0)}}}_{\geq 1 \text{ because } A_{F_{\text{others}}}(0) \geq A_{F_{\text{wg}}}(0)} - \underbrace{\frac{F_{\text{wg}}(0)}{1 - F_{\text{wg}}(0)}}_{<1 \text{ because } A_{F_{\text{wg}}}(0) > 0.5}\right)$$

$$= C(f_{\text{others}}(0)F_{\text{wg}}(0) - f_{\text{wg}}(0)F_{\text{others}}(0)) \tag{137}$$

where (133) follows from Lemma 10. $\qquad\square$

### E.2.1 NECESSARY CONDITION FOR OUTPERFORMING THE GROUP-AGNOSTIC REFERENCE

**Proposition 3** (Necessary condition for outperforming the group-agnostic reference)**.** *Assume that $1/2 < A_{F_{\text{wg}}}(0) < A_{F_{\text{others}}}(0) < 1$ and the worst-group density $f_{\text{wg}}(0) > 0$. If $\tilde{A}_{F_{\text{wg}}}(\tau) \leq A_{F_{\text{wg}}}(\tau)$ for all $\tau \geq 0$, then*

$$\frac{f_{\text{others}}(0)}{f_{\text{wg}}(0)} \leq \frac{1 - A_{F_{\text{others}}}(0)}{1 - A_{F_{\text{wg}}}(0)}. \tag{5}$$

*Proof.* Recall that

$$A_{F_{\text{wg}}}(\tau) = \frac{1}{1 + \frac{\mathbf{IF}_{\text{wg}}(\tau)}{\mathbf{CF}_{\text{wg}}(\tau)} \cdot \frac{I(\tau)}{C(\tau)}} \tag{138}$$

$$\tilde{A}_{F_{\text{wg}}}(\tau) = \frac{1}{1 + \frac{\mathbf{IF}_{\text{wg}}(0)}{\mathbf{CF}_{\text{wg}}(0)} \cdot \frac{I(\tau)}{C(\tau)}}. \tag{139}$$

If $A_{F_{wg}}(\tau) \geq \tilde{A}_{F_{wg}}(\tau)$ for all $\tau \geq 0$, then

$$\frac{d}{d\tau} \frac{\mathbf{IF}_{wg}(\tau)}{\mathbf{CF}_{wg}(\tau)}\bigg|_{\tau=0} \leq 0. \tag{140}$$

From Lemma 11,

$$C(f_{others}(0)F_{wg}(0) - f_{wg}(0)F_{others}(0)) \leq \frac{d}{d\tau} \frac{\mathbf{IF}_{wg}(\tau)}{\mathbf{CF}_{wg}(\tau)}\bigg|_{\tau=0} \tag{141}$$

for some positive constant $C$. Combined,

$$f_{others}(0)F_{wg}(0) - f_{wg}(0)F_{others}(0) \leq 0. \tag{142}$$

$\square$

### E.2.2 OUTPERFORMING THE GROUP-AGNOSTIC REFERENCE REQUIRES SMALLER SCALING FOR LOG-CONCAVE DISTRIBUTIONS

**Corollary 1** (Outperforming the group-agnostic reference requires smaller scaling for log-concave distributions). *Assume that $1/2 < A_{F_{wg}}(0) < A_{F_{others}}(0) < 1$, $F_{wg}$ is log-concave, and $f_{others}(\tau) = vf_{wg}(v(\tau - \mu_{others}) + \mu_{wg})$ for all $\tau \in \mathbb{R}$, where $v$ is a scaling factor. If $\tilde{A}_{F_{wg}}(\tau) \leq A_{F_{wg}}(\tau)$ for all $\tau \geq 0$, $v < 1$.*

*Proof.* By Proposition 3, if $A_{F_{wg}}(\tau) \geq \tilde{A}_{F_{wg}}(\tau)$ for all $\tau \geq 0$, then:

$$f_{others}(0)F_{wg}(0) - f_{wg}(0)F_{others}(0) \leq 0, \tag{143}$$

and equivalently,

$$\frac{f_{others}(0)}{F_{others}(0)} \leq \frac{f_{wg}(0)}{F_{wg}(0)}. \tag{144}$$

From the definition of $f_{others}$,

$$v \leq \frac{f_{wg}(0)/F_{wg}(0)}{f_{wg}(-\mu_{others}v + \mu_{wg})/F_{wg}(-\mu_{others}v + \mu_{wg})} \tag{145}$$

Because $A_{F_{others}}(0) > A_{F_{wg}}(0)$, $-\mu_{others}v + \mu_{wg} < 0$. Applying log-concavity yields

$$v < 1. \tag{146}$$

Thus, when $v > 1$, there exists some threshold $\tau \geq 0$ where $\tilde{A}_{F_{wg}}(\tau) > A_{F_{wg}}(\tau)$. $\square$

### E.3 TRANSLATED, LOG-CONCAVE DISTRIBUTIONS ALWAYS UNDERPERFORM THE GROUP-AGNOSTIC REFERENCE

We now present a proof of Proposition 4. Assume $f_{wg}$ and $f_{others}$ are log concave and symmetric, the worst group has mean $\mu_{wg}$, the combination of other group(s) has mean $\mu_{others}$, and $f_{others}(x) = f_{wg}(x - (\mu_{others} - \mu_{wg}))$, (the densities are translations of each other.) For convenience, define $d = \mu_{others} - \mu_{wg}$. This implies:

$$F_{others}(x) = F_{wg}(x - (\mu_{others} - \mu_{wg})) \tag{147}$$
$$= F_{wg}(x - d). \tag{148}$$

We will first show that wg is indeed the worst group at full coverage; i.e. $A_{F_{wg}}(0) < A_{F_{others}}(0)$:

**Lemma 12.** *If $f_{wg}$ is a PDF and $f_{others}$ is obtained by translating $f_{wg}$ to the right (i.e. $f_{others}(x) = f_{wg}(x - d)$ for $d > 0$), and $F_{wg}$ and $F_{others}$ are their associated CDFs, $A_{F_{wg}}(0) < A_{F_{others}}(0)$.*

*Proof.* We have:

$$A_{F_{wg}}(0) = 1 - F_{wg}(0) \tag{149}$$
$$\leq 1 - F_{wg}(-d) \tag{150}$$
$$= 1 - F_{others}(0) \tag{151}$$
$$= A_{F_{others}}(0). \tag{152}$$

$\square$

We next show that $\mathbf{CF}_{\mathrm{wg}}(\tau)$ is monotonically decreasing in $\tau$:

**Lemma 13.** *If $f_{\mathrm{wg}}$, $F_{\mathrm{wg}}$, $f_{\mathrm{others}}$, $F_{\mathrm{others}}$, $p$, and $\mathbf{CF}_{\mathrm{wg}}$ are as described, $\mathbf{CF}_{\mathrm{wg}}$ is monotonically decreasing in $\tau$.*

*Proof.* First, defining $\mathbf{CF}_{\mathrm{wg}}$ in terms of $F_{\mathrm{wg}}$, we have:

$$\mathbf{CF}_{\mathrm{wg}}(\tau) = \frac{p(1 - F_{\mathrm{wg}}(\tau))}{p(1 - F_{\mathrm{wg}}(\tau)) + (1 - p)(1 - F_{\mathrm{others}}(\tau))} \tag{153}$$

$$= \frac{p(1 - F_{\mathrm{wg}}(\tau))}{p(1 - F_{\mathrm{wg}}(\tau)) + (1 - p)(1 - F_{\mathrm{wg}}(\tau - d))} \tag{154}$$

$$= \frac{pF_{\mathrm{wg}}(2\mu_{\mathrm{wg}} - \tau)}{pF_{\mathrm{wg}}(2\mu_{\mathrm{wg}} - \tau) + (1 - p)F_{\mathrm{wg}}(2\mu_{\mathrm{wg}} - \tau + d)} \tag{155}$$

$$= \frac{1}{1 + \frac{(1-p)F_{\mathrm{wg}}(2\mu_{\mathrm{wg}} - \tau + d)}{pF_{\mathrm{wg}}(2\mu_{\mathrm{wg}} - \tau)}}. \tag{156}$$

Taking the derivative of $\mathbf{CF}_{\mathrm{wg}}$, we have:

$$\frac{d}{d\tau}\mathbf{CF}_{\mathrm{wg}}(\tau) = \frac{d}{d\tau}\left(\frac{1}{1 + \frac{(1-p)F_{\mathrm{wg}}(2\mu_{\mathrm{wg}} - \tau + d)}{pF_{\mathrm{wg}}(2\mu_{\mathrm{wg}} - \tau)}}\right) \tag{157}$$

$$= -\left(\frac{1}{1 + \frac{(1-p)F_{\mathrm{wg}}(2\mu_{\mathrm{wg}} - \tau + d)}{pF_{\mathrm{wg}}(2\mu_{\mathrm{wg}} - \tau)}}\right)^2 \left(\frac{1 - p}{p}\right)\frac{d}{d\tau}\left(\frac{F_{\mathrm{wg}}(2\mu_{\mathrm{wg}} - \tau + d)}{F_{\mathrm{wg}}(2\mu_{\mathrm{wg}} - \tau)}\right) \tag{158}$$

$$= -c\left(-f_{\mathrm{wg}}(2\mu_{\mathrm{wg}} - \tau + d)F_{\mathrm{wg}}(2\mu_{\mathrm{wg}} - \tau) + f_{\mathrm{wg}}(2\mu_{\mathrm{wg}} - \tau)F_{\mathrm{wg}}(2\mu_{\mathrm{wg}} - \tau + d)\right), \tag{159}$$

where $c$ is a positive constant (since $p$ and $1 - p$ are positive and all of the terms are squares.) Thus, we can say that $\mathbf{CF}_{\mathrm{wg}}(\tau)$ is monotonically decreasing for all $\tau$ if:

$$f_{\mathrm{wg}}(2\mu_{\mathrm{wg}} - \tau + d)F_{\mathrm{wg}}(2\mu_{\mathrm{wg}} - \tau) \le f_{\mathrm{wg}}(2\mu_{\mathrm{wg}} - \tau)F_{\mathrm{wg}}(2\mu_{\mathrm{wg}} - \tau + d) \tag{160}$$

$$\frac{f_{\mathrm{wg}}(2\mu_{\mathrm{wg}} - \tau + d)}{F_{\mathrm{wg}}(2\mu_{\mathrm{wg}} - \tau + d)} \le \frac{f_{\mathrm{wg}}(2\mu_{\mathrm{wg}} - \tau)}{F_{\mathrm{wg}}(2\mu_{\mathrm{wg}} - \tau)}. \tag{161}$$

Now, since $d > 0$, this follows from the log concavity of $f_{\mathrm{wg}}$. Therefore, the fraction of correct examples that come from group 1 is a decreasing function of $\tau$. □

Next, we show that the $\mathbf{IF}_{\mathrm{wg}}(\tau)$ is monotonically increasing in $\tau$:

**Lemma 14.** *If $f_{\mathrm{wg}}$, $F_{\mathrm{wg}}$, $f_{\mathrm{others}}$, $F_{\mathrm{others}}$, $p$ and $\mathbf{IF}_{\mathrm{wg}}$ are as described, $\mathbf{IF}_{\mathrm{wg}}$ is monotonically increasing in $\tau$.*

*Proof.* First, we note that:

$$\mathbf{IF}_{\mathrm{wg}}(\tau) = \frac{pF_{\mathrm{wg}}(-\tau)}{pF_{\mathrm{wg}}(-\tau) + (1 - p)F_{\mathrm{wg}}(-\tau - d)} \tag{162}$$

$$= \frac{1}{1 + \frac{(1-p)F_{\mathrm{wg}}(-\tau - d)}{pF_{\mathrm{wg}}(-\tau)}}. \tag{163}$$

We will show that the derivative of $\mathbf{IF}_{\mathrm{wg}}(\tau)$ with respect to $\tau$ is positive. Doing so gives us:

$$\frac{d}{d\tau}\mathbf{IF}_{\mathrm{wg}}(\tau) = \frac{d}{d\tau}\frac{1}{1 + \frac{(1-p)F_{\mathrm{wg}}(-\tau - d)}{pF_{\mathrm{wg}}(-\tau)}} \tag{164}$$

$$= -\left(\frac{1}{1 + \frac{(1-p)F_{\mathrm{wg}}(-\tau - d)}{pF_{\mathrm{wg}}(-\tau)}}\right)^2 \left(\frac{1 - p}{p}\right)\frac{d}{d\tau}\left(\frac{F_{\mathrm{wg}}(-\tau - d)}{F_{\mathrm{wg}}(-\tau)}\right) \tag{165}$$

$$= -c\left(-f_{\mathrm{wg}}(-\tau - d)F_{\mathrm{wg}}(-\tau) + f_{\mathrm{wg}}(-\tau)F_{\mathrm{wg}}(-\tau - d)\right), \tag{166}$$

where $c$ is positive since it is the product of squared terms and $(1 - p)/p$, which is also positive. Thus, $\frac{d}{d\tau} \mathbf{IF}_{\text{wg}}(\tau) \geq 0$ is equivalent to:

$$- (-f_{\text{wg}}(-\tau - d)F_{\text{wg}}(-\tau) + f_{\text{wg}}(-\tau)F_{\text{wg}}(-\tau - d)) \geq 0 \tag{167}$$

$$f_{\text{wg}}(-\tau - d)F_{\text{wg}}(-\tau) \geq f_{\text{wg}}(-\tau)F_{\text{wg}}(-\tau - d) \tag{168}$$

$$\frac{f_{\text{wg}}(-\tau - d)}{F_{\text{wg}}(-\tau - d)} \geq \frac{f_{\text{wg}}(-\tau)}{F_{\text{wg}}(-\tau)}. \tag{169}$$

Since $f_{\text{wg}}$ is log-concave and $d$ is positive, the last inequality is true, so $\mathbf{IF}_{\text{wg}}(\tau)$ is monotonically increasing in $\tau$, as desired. $\qquad \square$

Lastly, we show that Lemma 13 and Lemma 14 imply that the ratio $\frac{\mathbf{IF}_{\text{wg}}(\tau)}{\mathbf{CF}_{\text{wg}}(\tau)}$ is monotonically increasing in $\tau$.

**Lemma 15.** $\frac{\mathbf{IF}_{\text{wg}}(\tau)}{\mathbf{CF}_{\text{wg}}(\tau)}$ *is monotonically increasing in* $\tau$

*Proof.* We simply follow the quotient rule:

$$\frac{d}{d\tau} \frac{\mathbf{IF}_{\text{wg}}(\tau)}{\mathbf{CF}_{\text{wg}}(\tau)} = \mathbf{CF}_{\text{wg}}(\tau)IF_1'(\tau) - \mathbf{IF}_{\text{wg}}(\tau)CF_1'(\tau) \tag{170}$$

$$\geq 0, \tag{171}$$

where we note that $\mathbf{CF}_{\text{wg}}(\tau), \mathbf{IF}_{\text{wg}}(\tau) \geq 0$, $CF_1'(\tau) < 0$ from Lemma 13, and $IF_1'(\tau) > 0$ from Lemma 14. $\qquad \square$

We will now prove the main result of this section:

### E.3.1 TRANSLATED, LOG-CONCAVE DISTRIBUTIONS ALWAYS UNDERPERFORM THE GROUP-AGNOSTIC REFERENCE

**Proposition 4** (Translated log-concave distributions underperform the group-agnostic reference)**.** *Assume $F_{\text{wg}}$ and $F_{\text{others}}$ are log-concave and $f_{\text{others}}(\tau) = f_{\text{wg}}(\tau - d)$ for all $\tau \in \mathbb{R}$. Then for all $\tau \geq 0$,*

$$A_{F_{\text{wg}}}(\tau) \leq \tilde{A}_{F_{\text{wg}}}(\tau). \tag{6}$$

*Proof.* We consider the case of underperforming the group-agnostic reference: the case of overperforming the group-agnostic reference is analogous. Assume the worst group is the lower accuracy group (so $d$ is positive.) Using the definition of selective accuracy, we have:

$$A_{F_{\text{wg}}}(\tau) = \frac{C_{\text{wg}}(\tau)}{C_{\text{wg}}(\tau) + I_1(\tau)} \tag{172}$$

$$= \frac{\mathbf{CF}_{\text{wg}}(\tau)C(\tau)}{\mathbf{CF}_{\text{wg}}(\tau)C(\tau) + \mathbf{IF}_{\text{wg}}(\tau)I(\tau)} \tag{173}$$

$$= \frac{1}{1 + \frac{\mathbf{IF}_{\text{wg}}(\tau)}{\mathbf{CF}_{\text{wg}}(\tau)} * \frac{I(\tau)}{C(\tau)}} \tag{174}$$

$$\leq \frac{1}{1 + \frac{\mathbf{IF}_{\text{wg}}(0)}{\mathbf{CF}_{\text{wg}}(0)} * \frac{I(\tau)}{C(\tau)}} \qquad \text{[by Lemma 15]} \tag{175}$$

$$= \frac{\mathbf{CF}_{\text{wg}}(0)C(\tau)}{\mathbf{CF}_{\text{wg}}(0)C(\tau) + \mathbf{IF}_{\text{wg}}(0)I(\tau)} \tag{176}$$

$$= \tilde{A}_{F_{\text{wg}}}(\tau). \tag{177}$$

Thus, the worst group underperforms the group-agnostic reference, as desired. $\qquad \square$

## F    SIMULATIONS

To demonstrate that it is possible but challenging to outperform the group-agnostic reference, we present simulation results on margin distributions that are mixtures of two Gaussians. Following the setup from Section 6, we consider a best-group margin distribution $\mathcal{N}(1, 1)$ and a worst-group margin distribution $\mathcal{N}(\mu, \sigma^2)$ varying the parameters $\mu, \sigma$. We set the fraction of mass from the worst group, $p$ to be 0.5. We evaluate whether selective classification outperforms the group-agnostic reference, plotting the results in Figure 9. We observe that selective classification outperforms the group-agnostic reference under some parameter settings, notably when the necessary condition in Proposition 3 is met and when the variance is smaller as implied by Corollary 1, but it underperforms the group-agnostic reference under most parameter settings.

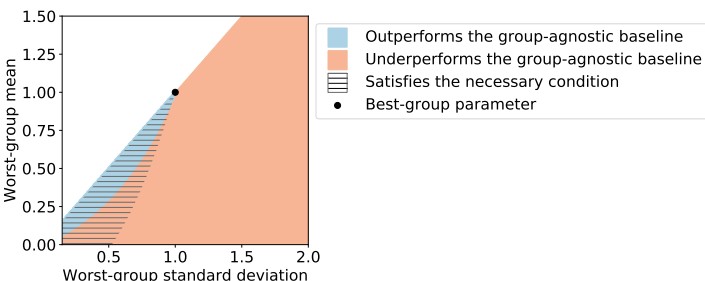

Figure 9: Simulation results on margin distributions that are mixtures of two Gaussians, where we vary the mean and the variance of the worst-group margin distributions. For parameters corresponding to the blue and orange regions, the worst group outperforms and underperforms the group-agnostic reference, respectively. We do not consider parameters in the white region to maintain that they yield full-coverage accuracies that are worse than the other group. We shade parameters that satisfy the necessary condition in Proposition 3.

In addition to whether the worst group underperforms or outperforms the group-agnostic reference, we study the magnitude of the difference in the worst-group accuracy with respect to the group-agnostic reference through the same simulations (Figure 10). Concretely, we compute the difference in worst-group accuracy with respect to the group-agnostic reference, taking the worst-case difference across thresholds: $\max_\tau \tilde{A}_{F_{\mathrm{wg}}}(\tau) - A_{F_{\mathrm{wg}}}(\tau)$. We observe significant disparities between the observed worst-group accuracy and the group-agnostic reference.

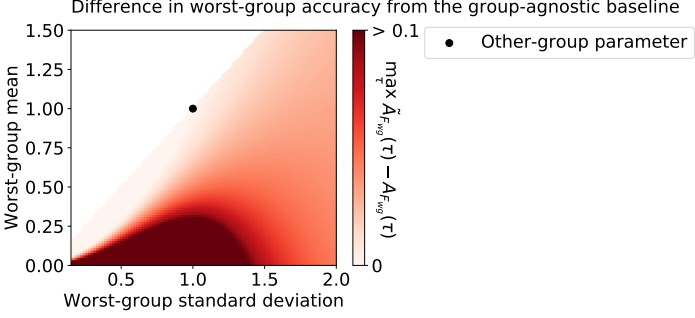

Figure 10: Simulation results on margin distributions that are mixtures of two Gaussians, where we vary the mean and the variance of the worst-group margin distribution, and keep the other group's margin distribution fixed with mean and variance 1. We compute the difference in worst-group accuracy with respect to the group-agnostic reference, taking the worst-case difference across thresholds: $\max_\tau \tilde{A}_{F_{\mathrm{wg}}}(\tau) - A_{F_{\mathrm{wg}}}(\tau)$.

Lastly, we simulate the effects of varying the worst-group and other-group means while keeping the variance fixed at $\sigma^2 = 1$ for both groups, following the same simulation protocol otherwise.

We similarly observe substantial gaps between the observed worst-group accuracy and the group-agnostic reference (Figure 11).

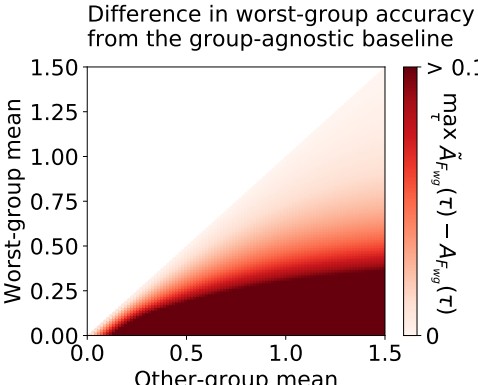

Figure 11: Simulation results on margin distributions that are mixtures of two Gaussians, where we vary the means of the two margin distributions, keeping their variances fixed at 1. We compute the difference in worst-group accuracy with respect to the group-agnostic reference, taking the worst-case difference across thresholds: $\max_\tau \tilde{A}_{F_{\text{wg}}}(\tau) - A_{F_{\text{wg}}}(\tau)$.

