# OpenReview forum: "Selective Classification Can Magnify Disparities Across Groups"
_ICLR.cc/2021/Conference — ICLR 2021 Poster_

### Official Review · AnonReviewer2 · 2020-10-13
**Interesting paper about potential biases induced by abstention**

**Rating:** 7
**Confidence:** 4

**Review:**

This paper discusses the impact of classifier abstention on the performance obtained for different groups of data. In particular, the authors show that, while abstaining in case of uncertainty in general leads to an improvement in predictive performance, it can also lead to worse results for specific groups. In a somewhat atypical paper structure, the authors first illustrate this empirically on five banchmark datasets. Later in the paper, they also present theoretical results that explain the empirical results.

This is an interesting paper about an original research topic. The take-home message is not so surprising, but the analysis is thorough. I particularly appreciated that the empirical results are also supported by theoretical results. I didn't do a detailed check of the proofs, but I don't see a reason why the propositions should not hold.

Overall the paper is also well written and I enjoyed reading it. A few things are not very clear to me:

1. In Section 3 it is briefly mentioned that two scores are considered as a representation of uncertainty: softmax and a score based on MC dropout. For the latter it is not clear to me how the score is computed. Is this the variance over different predictions for an instance? Moreover, in Figures 2 and 4 it is not mentioned which of the two scores is used to obtain the graphs. I would have expected to see the results for softmax in the main paper, since this is most often used as uncertainty score, but the horizontal axis in Figure 4 is not limited to the interval [-1,1], so this cannot be softmax. This needs to be explained better.

2. Related to the previous remark, I would also argue that the margin distributions depend very strongly on the uncertainty measure that is used. As discussed e.g. in the work of Kendall and Gal, softmax looks at aleatoric uncertainty, whereas MC dropout  looks at epistemic uncertainty. Furthermore, also a lot of other uncertainty measures exist in the literature. For example, in multi-class classification, one often constructs models that abstain on instances that are very dissimilar to the training data (so-called out-of-distribution samples). Thus, different measures represent different types of uncertainty, so I also expect to see very different margin distributions on specific datasets.

3.  In the description of the experimental setup, I found the notion "spurious attribute" unclear. For "spurious correlation" it is clear what is meant, but I have never heard of a spurious attribute, and I also don't know what that would be. Aren't you just referring to a confounding attribute, as commonly used in the statistical literature? Does it make a difference for the analysis whether the group variable is observed or unobserved? It might be good to elaborate on this.

4. It is also not very clear what the practical usefulness of the obtained results is. It is nice to know that the performance can drop for specific groups when abstention is allowed, but this is not necessarily a bad thing. In practice one encounters a lot of situations where machine learning models are used as a prescreening tool, followed by a manual check for cases that are hard to classify by models. In such situations one could decide to use a particular algorithm only for specific groups, whereas badly-scoring groups would deserve a special treatment, such as a manual screening.

---

> ### Author Response · Authors · 2020-11-24
> **Response to Reviewer #2**
>
> We thank R2 for their helpful feedback and are glad they enjoyed reading the paper.
>
> **Q1: “In Figures 2 and 4 it is not mentioned which of the two scores is used to obtain the graphs. I would have expected to see the results for softmax in the main paper, since this is most often used as uncertainty score, but the horizontal axis in Figure 4 is not limited to the interval [-1,1], so this cannot be softmax.”**
>
> We apologize for the confusion, and have updated our setup section to explicitly clarify this. The results in Figures 2 and 4 are indeed for softmax-response selective classifiers. However, instead of taking the maximum softmax probability (which would result in confidences in [-1, 1], as R2 suggests), we instead take (a normalized version of) the logit corresponding to the maximum softmax probability. This does not change the accuracy-coverage curve, as the softmax probability is a monotone function of its corresponding normalized logit. However, it makes the distribution of margins easier to interpret, as the resulting distribution tends to be Gaussian (for example, [1] gives conditions under which the distribution of logits is Gaussian).
>
>
> [1] K. Balasubramanian, P. Donmez, and G. Lebanon. Unsupervised supervised learning II: Margin-based classification without labels. Journal of Machine Learning Research (JMLR), 12:3119– 3145, 2011.
>
> ---
> **Q2: “In Section 3 it is briefly mentioned that two scores are considered as a representation of uncertainty: softmax and a score based on MC dropout. For the latter it is not clear to me how the score is computed. Is this the variance over different predictions for an instance?”**
>
> We follow the procedure in the original paper that introduced MC dropout as a form of uncertainty quantification [2]. Specifically, given a model with a dropout layer, we first use standard forward propagation (without dropout) to obtain a prediction. We then query the model multiple times with dropout activated to get various softmax outputs, and define the confidence to be the negative variance of the softmax probability of the class corresponding to our original prediction. We detail this in Appendix B.1, and have added a note to the setup linking to it. Just to clarify, all of the experimental figures in the main paper are with softmax-response selective classifiers, and the MC-dropout results (which show similar trends) are all in the Appendix.
>
> [2] Y. Gal and Z. Ghahramani. Dropout as a Bayesian approximation: Representing model uncertainty in deep learning. In International Conference on Machine Learning (ICML), 2016.
>
> ---
> **Q3: “Related to the previous remark, I would also argue that the margin distributions depend very strongly on the uncertainty measure that is used. As discussed e.g. in the work of Kendall and Gal, softmax looks at aleatoric uncertainty, whereas MC dropout looks at epistemic uncertainty.”**
>
> Thank you for bringing this up. We agree with R2 that the margin distribution depends strongly on the chosen uncertainty measure. The fact that the same behavior manifests on both softmax response and MC-dropout suggests that the problem of group disparities is a broader problem, but studying the extent to which this other existing selective classification methods are affected (and whether any particular methods of uncertainty measurement mitigate the group disparity problem) would be a good direction for future work. We have added a note about this to the Discussion section.
>
> ---
> **Q4: “In the description of the experimental setup, I found the notion "spurious attribute" unclear. For "spurious correlation" it is clear what is meant, but I have never heard of a spurious attribute, and I also don't know what that would be. Aren't you just referring to a confounding attribute, as commonly used in the statistical literature? Does it make a difference for the analysis whether the group variable is observed or unobserved? It might be good to elaborate on this.”**
>
> We used “spurious attribute” as short-hand for “attribute that is spuriously correlated with the label”. We apologize for the confusion and have clarified this in the text. We have also made it clearer in the setup that in this paper, we assume the group variable is unobserved at test time. If the group variable is observed at both training and test time, then it can essentially be counted as part of the input features, and one could hope that the resulting models on this richer feature set would have smaller group disparities and better uncertainty estimates (though this is not guaranteed). The analysis of the margin distributions does not depend on whether the group variable is observed or unobserved, so all of it will still go through, i.e., if the worst-group accuracy is significantly worse than the average accuracy at full coverage, then selective classification is likely to magnify this disparity. Thank you for bringing this up!

---

> > ### Author Response · Authors · 2020-11-24
> > **Response to Reviewer #2 (continued)**
> >
> > **Q5: “It is also not very clear what the practical usefulness of the obtained results is. It is nice to know that the performance can drop for specific groups when abstention is allowed, but this is not necessarily a bad thing. In practice one encounters a lot of situations where machine learning models are used as a prescreening tool, followed by a manual check for cases that are hard to classify by models. In such situations one could decide to use a particular algorithm only for specific groups, whereas badly-scoring groups would deserve a special treatment, such as a manual screening.”**
> >
> > We agree with R2 that there are many different settings in which selective classifiers can be used. In our settings, we have assumed that the group identities are unknown or otherwise hard to obtain at test time, precluding the possibility that one could opt to deploy different algorithms for different group (for example, the group could be a particular disease subgroup or co-morbidity, whose annotation might be as difficult to obtain as the target label). However, if it is possible to reliably estimate group identities at test time, then as R2 points out, one could not deploy the model for those groups. The feasibility and impact of doing so depends on the particular setting (e.g., if the decision is time-sensitive, then the cost of abstention might be high). For simplicity, in this paper we have focused on the selective classifier in isolation, but we agree that a broader study of group disparities in selective classifiers that operate one component of a larger system would be an important direction for future work. We have added this to the discussion -- thank you!

---

### Official Review · AnonReviewer4 · 2020-10-28
**Selective Classification Can Magnify Disparities Across Groups**

**Rating:** 8
**Confidence:** 4

**Review:**

Summary: This paper demonstrates and analyzes the accuracy disparities induced by selective classification on subgroups. It evaluates these induced disparities across five datasets from different domains, and provides a theoretical analysis of the conditions under which selective classification may decrease accuracy or magnify disparities via evaluation of the margin distribution.

Overall, this paper is a useful addition to a growing literature on the disparate impact of various machine learning techniques on subgroups. The analysis in terms of margin distribution is both appropriate to the selective classification task, and novel, as far as I am aware. The theoretical analysis clearly describes conditions under which the various observed experimental results may generally hold. The (short) section on DRO also suggests a promising way out of this problem, informed by the results and analysis in the paper. The paper could use some minor edits and clarifications, but I believe it would be a valuable addition to the conference.

Note: I did not evaluate or check all of the proofs.

#### Major Comments:

* I found the description of the baselines confusing; it is not made clear that C and I are being defined (I had to scan to see if they were previously discussed) and a great deal of text is used to describe a baseline construction procedure which could be encapsulated in a simple algorithm (my recommendation) or set of numbered steps.

* Throughout, the paper uses a definition of "average accuracy" (defined in Sec. 3) which is not actually average accuracy -- it is average accuracy on covered points. I find this to be a bit misleading, or at the very least a confusing overloading of the term, and would recommend introducing a new name for this metric and re-labeling the y-axis of the plots that use it.

* The DRO section (7) feels like an afterthought, but it is actually quite important to the paper. There are several recent works which identify disparate model performance impacts on subgroups (e.g. due to model compression, differential privacy, etc.), with no clear solution. The connection to DRO suggests a potential solution, at least in practice, for the problem identified in this paper. It would be nice to, at the very least, see some of the discussion of DRO upgraded from the appendix to the main text, if not a more thorough analysis.

#### Minor comments:

* I did not feel that the actual process of training a selective classifier (vs. a standard classifier) was clearly explained; since this is fundamental to the analysis, it would be useful to have this clearly signposted for readers unfamiliar with selective classification (such as this reviewer).

#### Typos etc.

Sec. 7: "to their average accuracies Figure 5"

####

UPDATE: I have reviewed the author response and the revisions. All of my main concerns were addressed. I believe that these improved the paper further, and while they do not change my rating, the paper is still a clear "accept" in my view.

---

> ### Author Response · Authors · 2020-11-24
> **Response to Reviewer #4**
>
> We thank R4 for their helpful feedback and constructive suggestions on many aspects of the paper.
>
> **Q1: “I found the description of the baselines confusing; it is not made clear that C and I are being defined (I had to scan to see if they were previously discussed) and a great deal of text is used to describe a baseline construction procedure which could be encapsulated in a simple algorithm (my recommendation) or set of numbered steps.”**
>
> Thank you for the good suggestion, and we apologize for the initial confusion. We followed R4’s recommendation and have added algorithm boxes to describe the baselines, which we hope is easier to parse.
>
> ---
> **Q2: “Throughout, the paper uses a definition of "average accuracy" (defined in Sec. 3) which is not actually average accuracy -- it is average accuracy on covered points.”**
>
> Thanks for raising this point. The literature on selective classification is not completely consistent, but the term “accuracy” is commonly used to refer to selective accuracy (i.e., accuracy on the covered points), presumably to be concise. We have explicitly clarified this in the setup, and have also added “(selective)” before the first mention of accuracy in each section to further emphasize the equivalence. As R4 suggested, we have also relabeled the y-axes of the corresponding plots.
>
> ---
> **Q3: “The DRO section (7) feels like an afterthought, but it is actually quite important to the paper…”**
>
> We agree with R4. In response, we have emphasized the DRO section in the introduction; reinforced its connection with the analysis; and expanded the existing DRO section, following R4’s suggestion of moving some of the results and discussion from the appendix up to the main text. In particular, we now highlight the surprising result that the average and worst-group distributions of margins are (qualitatively) strikingly similar with group DRO models. Thanks for bringing this up.
>
> ---
> **Q4: “I did not feel that the actual process of training a selective classifier (vs. a standard classifier) was clearly explained; since this is fundamental to the analysis, it would be useful to have this clearly signposted for readers unfamiliar with selective classification (such as this reviewer).”**
>
> We apologize for the lack of clarity there. We have expanded the setup to explicitly discuss training as well as how we obtain the confidence function for softmax response (bringing it up from the appendix). We note that the analysis should apply generally to any confidence-based selective classifier $(\hat{y}, \hat{c})$ regardless of how it was trained.

---

### Official Review · AnonReviewer1 · 2020-10-29
**Review of Paper2955**

**Rating:** 7
**Confidence:** 3

**Review:**

The paper draws attention to a problem in the selective classification. In particular, it implies that selective classifiers can perform unnegligible accuracy disparities across various groups. To capture such information, the authors theoretically analyze the margin distributions and compare to the group-agnostic baseline, which can help to explain the accuracy disparities shown in the training result of 5 datasets.

Overall, I like the idea of theoretically analyzing the accuracy-coverage curve by margin distribution and I vote for accepting.

Pros:
1. The paper takes an important issue of selective classification: accuracy disparities between groups. For me, it's possibly related to ''fairness'' in classification, which we shall treat carefully in real-world problems.

2. The proposed margin distribution for analysis is reasonable. The theoretical analysis is helpful to explain the observation in experiments.

3. This paper provides experiments to illustrate their findings in typical datasets and defend their analysis.

Below are my major concerns. Hopefully the authors can address them in the rebuttal period.

Cons:
1. Although exploiting the margin distribution can provide good reasoning on the monotonicity of accuracy-coverage curves, analysis of comparison to group-agnostic baseline seems not very clear to me:

(1) In that section, the authors consider the case of a mixture of two groups. Is this case typical enough to demonstrate the issues the authors want to imply?

(2) In proposition 5, the authors show that log-concave distributions always underperform the baseline. However, from my view, the importance of this issue lies on that the disparity is unnegligible, so a more refined estimate of the difference between accuracies, namely, $|A_F(\tau)-\tilde A_F(\tau)|$ (may need some extra conditions) will be more helpful to explain results on CheXpert-device.


Some typos:
1. Page 4: there are extra "]" in both C(tau) and I(tau).

---

> ### Author Response · Authors · 2020-11-24
> **Response to Reviewer #1**
>
> We thank R1 for their helpful feedback. R1’s questions were primarily around the analysis of the group-agnostic baseline (Section 6):
>
> **Q1: “In that section, the authors consider the case of a mixture of two groups. Is this case typical enough to demonstrate the issues the authors want to imply?”**
>
> Thank you for pointing this out; it is a great question. The two-group case is indeed general enough to demonstrate the issues we wanted to highlight; since we focus on worst-group accuracy, we can reduce a problem with $n$ groups into a two-group problem by treating one group as the worst group and the other group as the remaining $n-1$ groups combined. We have revised the manuscript to reflect this.
>
> ---
> **Q2: “In proposition 5, the authors show that log-concave distributions always underperform the baseline. However, from my view, the importance of this issue lies on that the disparity is unnegligible, so a more refined estimate of the difference between accuracies, namely, $|A_F(\tau) - \tilde{A}_F(\tau)|$, (may need some extra conditions) will be more helpful to explain results on CheXpert-device.”**
>
> We agree with R1 that the magnitude of the difference $|A_F(\tau) - \tilde{A}_F(\tau)|$ is important. Unfortunately, as R1 points out, the magnitude depends on a lot of factors that would require specific assumptions on the parametric form of the margin distributions (e.g., assuming they are Gaussian with a particular standard deviation). As such, we decided to go with a smaller set of assumptions (i.e., just log-concavity) so that our results in Proposition 5 would apply more generally. While it does not provide for quantification, we believe that this result is still important, as matching the group-agnostic baseline on the worst group is in some sense the minimum that we should hope to achieve, and it is therefore useful to know the conditions under which we cannot even match it (even if the difference is small).
>
> To give readers a better sense of the rate at which the difference from the group-agnostic baseline grows, in Appendix F, we have now added numerical simulations of this difference for particular parametric assumptions (i. a mixture of two Gaussians with a fixed variance, varying the means; ii. a mixture of two Gaussians with one fixed distribution, varying the mean and variance of the other distribution). Thank you for bringing this up.
>
> ---
> We have also fixed the typo that R1 pointed out; thank you!

---

### Official Review · AnonReviewer3 · 2020-10-29
**require some further effort**

**Rating:** 5
**Confidence:** 3

**Review:**

This paper mainly focuses on reporting a cautionary finding on using selective classification, but didn’t provide any solution. That is, selective classification can magnify existing accuracy disparities between various groups within a population, especially when there are spurious correlations. The authors studied the margin distribution to get some understandings about the problem.

If the disparity across different groups is important, performing a uniform standard classification over the unified data is not a good idea in the first place. It seems the group disparity problem is inherent to the full coverage classification and should be addressed at the full coverage in the first place, e.g., by using group-DRO models.  This is consistent with their finding (which is intuitive) that selective classification can uniformly improve group accuracies on group-DRO models, because the problem is inherent to the classification model used.

Although the authors tried to analyze their findings using margin distribution under the concept of the left-log-concave distribution, which does not lead to any solution for solving the disparity problem over the worst-group, but remains an observation. Moreover, it is unclear whether all the phenomena of selective classification magnifying disparity can be captured in their concepts and analysis, since their findings are based on empirical observations over a few datasets.

Overall I feel the current observation report requires further effort. It would be natural to expect the analysis on observations can lead to a solution for the observed problem.

---

> ### Author Response · Authors · 2020-11-24
> **Response to Reviewer #3**
>
> We thank R3 for their helpful feedback. Overall, R3 has three concerns, which we respond to in turn.
>
> **Q1: “If the disparity across different groups is important, performing a uniform standard classification over the unified data is not a good idea in the first place.”**
>
> R3 points out that the problem of group disparities can be addressed by not training a standard classification model over the unified data. Instead, one could use models that have smaller group disparities at full coverage, such as group DRO models. We agree with R3 that, where possible, it is good to use models which do not have group disparities at full coverage. Unfortunately, this is not always possible: for example, group DRO requires group annotations at training time, which are not always available. Simply evaluating if there are group disparities also requires group annotations at test time. Indeed, in some applications, model developers might not be aware of the identities of the important groups to study (i.e., the ones on which the model has substantially worse performance).
>
> In practice, standard models that are trained to minimize the average error over all of the available training data are extremely widely used, even in cases where the disparities across different groups are important. Moreover, selective classification has been considered in those contexts as well (e.g., selective classifiers have been recently proposed for both the chest x-ray and toxicity detection examples that we discuss in the introduction and study in our experiments). To the best of our knowledge, our work is the first to observe that selective classification can magnify group disparities. By highlighting this potential pitfall, we hope that our work can encourage practitioners to consider disparities across different groups and, as R3 suggests, explore alternative methods that mitigate those disparities (e.g., acquiring group annotations to train group DRO models).
>
> ---
> **Q2: “Although the authors tried to analyze their findings using margin distribution under the concept of the left-log-concave distribution, which does not lead to any solution for solving the disparity problem over the worst-group, but remains an observation.”**
>
> We have edited the manuscript so that the link between our analysis of the margin distribution and the subsequent group DRO section is clearer. Specifically, Propositions 4 and 5 suggest that selective classification generally magnifies existing full-coverage disparities (compared to the group-agnostic baseline). This motivates the subsequent exploration of group DRO models that focus on mitigating full-coverage disparities, which we find to be empirically successful. As we note above and in our Introduction, standard classifiers with group disparities are widely used today, and we hope that our analysis motivates practitioners to consider group disparities in the context of selective classification, and to ameliorate them by, e.g., acquiring group annotations.
>
> ---
> **Q3: “It is unclear whether all the phenomena of selective classification magnifying disparity can be captured in their concepts and analysis, since their findings are based on empirical observations over a few datasets.”**
>
> We believe that our empirical observations are reflective of a broader trend, as we found them to hold over 5 datasets, including both image and text applications, as well as both softmax-response and MC dropout selective classifiers. However, we agree with R3 that our empirical observations do not necessarily capture the behavior of selective classification on all other datasets. This motivated our theoretical analysis in Sections 5 and 6, which shows that this phenomenon arises under fairly general conditions (such as when the worst-case margin distribution is shifted to the left of the average margin distribution, and both are log-concave), especially in light of prior work showing that log-concave margin distributions are typical [1, 2].
>
> We agree with R3, of course, that our experiments and analysis do not capture every aspect of selective classification, and we hope that our work opens up interesting avenues of research in that direction: for example, studying the conditions under which the margin distribution is expected to be left-log-concave.
>
> [1] K. Balasubramanian, P. Donmez, and G. Lebanon. Unsupervised supervised learning II: Margin- based classification without labels. Journal of Machine Learning Research (JMLR), 12:3119–3145, 2011.
>
> [2] B. Lakshminarayanan, A. Pritzel, and C. Blundell. Simple and scalable predictive uncertainty estimation using deep ensembles. In Advances in Neural Information Processing Systems (NeurIPS), 2017.

---

### Author Response · Authors · 2020-11-24
**Response to all reviewers**

We are grateful to all of the reviewers for their detailed feedback and helpful suggestions, which we have incorporated into the updated manuscript. We briefly summarize the main changes here:

1. In our setup (Section 3), R2 and R4 had questions about how the selective classifiers were trained; how their resulting margin distributions were computed; what the definition of a spurious attribute was; and the difference between accuracy and selective accuracy. In response, we have rewritten the setup to clarify all of these points, including bringing up the formal definition of the confidence function for softmax-response selective classifiers from the appendix, and discussing training more explicitly.


2. We have taken R4’s helpful suggestion to rewrite the description of the group-agnostic baseline (Section 4) in an algorithm box, which we hope clarifies its construction.


3. R1 asked about the connection between theoretical analysis of the group-agnostic baseline, which considered the case of a mixture of two groups, and our empirical results, which consider > 2 groups. In response, we have edited the text in that section to clarify that we can treat the worst group as a single group and the other $n - 1$ groups as another group, which allows the analysis to also apply to a multiple-group setting.


4. Finally, R4 pointed out that the group DRO model in Section 7 was presented as if it were an afterthought, despite its importance to the paper. R3, similarly, commented on whether the analysis could more clearly direct us to a solution. To address these points, we have expanded Section 7, following R4’s suggestion to promote some of the material on group DRO from the appendix. In particular, we now show the margin distributions for group DRO as well, so that we can draw reader attention to the perhaps surprising result that in group DRO models, the margin distributions of the worst groups are qualitatively very similar to those of the average distributions. We have also edited the text of Section 7 to further connect group DRO to our theoretical analysis.


We include more detail on these and other changes in the separate responses below to each reviewer.

---

### Decision · Program_Chairs · 2021-01-07
**Final Decision**

**Decision:**

Accept (Poster)

**Comment:**

This is an interesting paper discussing the impact of classifier abstention on the performance obtained for different groups of data. The reviewers are either very (scores of 8, 7 and 7) or moderately (score of 5) positive about the paper. The main concern is that the paper does not directly propose a solution for the discovered problems. Nevertheless, it can initiate interesting discussions and research around them.